# DP-LDMs: Differentially Private Latent Diffusion Models

**Michael Liu**[*]                                                                 *mfliu@cs.ubc.ca*
*Department of Computer Science*
*University of British Columbia*

**Saiyue Lyu**[*]                                                                 *saiyuel@cs.ubc.ca*
*Department of Computer Science*
*University of British Columbia*

**Margarita Vinaroz**[*][†]                                        *margarita.vinaroz@tuebingen.mpg.de*
*University of Tübingen*
*International Max Planck Research School for Intelligent Systems (IMPRS-IS)*

**Mijung Park**[‡]                                                                 *mijungp@cs.ubc.ca*
*Department of Computer Science*
*University of British Columbia*

**Reviewed on OpenReview:** *https://openreview.net/forum?id=AkdQ266kHj*

## Abstract

Diffusion models (DMs) are one of the most widely used generative models for producing high quality images. However, a flurry of recent papers points out that DMs are least private forms of image generators, by extracting a significant number of near-identical replicas of training images from DMs. Existing privacy-enhancing techniques for DMs, unfortunately, do not provide a good privacy-utility tradeoff. In this paper, we aim to improve the current state of DMs with differential privacy (DP) by adopting the *Latent* Diffusion Models (LDMs). LDMs are equipped with powerful pre-trained autoencoders that map the high-dimensional pixels into lower-dimensional latent representations, in which DMs are trained, yielding a more efficient and fast training of DMs. Rather than fine-tuning the entire LDMs, we fine-tune only the *attention* modules of LDMs with DP-SGD, reducing the number of trainable parameters by roughly 90% and achieving a better privacy-accuracy trade-off. Our approach allows us to generate realistic, high-dimensional images (256x256) conditioned on text prompts with DP guarantees. Our approach provides a promising direction for training more powerful, yet training-efficient differentially private DMs, producing high-quality DP images. Our code is available at https://github.com/ParkLabML/DP-LDM.

## 1 Introduction

A flurry of recent work highlights the tension between increasingly powerful diffusion models and data privacy, e.g., (Wu et al., 2023; Carlini et al., 2023; Tang et al., 2023; Hu & Pang, 2023; Duan et al., 2023; Matsumoto et al., 2023; Somepalli et al., 2023) among many. These papers commonly conclude that diffusion models are extremely good at memorizing training data, leaking more than twice as much training data as GANs (Carlini et al., 2023). This raises the question of how diffusion models should be responsibly deployed. Carlini et al. (2023) seek to use differential privacy as a remedy for the memorization problem, but their attempt was unsuccessful due to the scalability issues of differential privacy. Our paper provides a practical and scalable fine-tuning routine for diffusion models with differential privacy guarantees to avoid generating identical images to those in the private dataset.

---

[*]Equal contribution.
[†]This work was done during a research visit to the Department of Computer Science at the University of British Columbia.
[‡]Some part of this project was done at the Technical University of Denmark.

To tackle the privacy concerns, Dockhorn et al. (2023) propose to use DP-SGD (Abadi et al., 2016) when training DMs, creating a method called *DPDM*. However, DP trained DMs yield rather underwhelming performance when evaluated on datasets such as CIFAR10 and CelebA. Recently, Ghalebikesabi et al. (2023) proposed a method, which is often referred to as *DP-Diffusion*, to pre-train a large diffusion-based generator using public data, then fine-tune it with private data for a relatively small number of epochs using DP-SGD.

In this paper, our goal is to further improve the performance of DP-fine-tuned DMs. To achieve this, we build on *latent diffusion models (LDMs)* (Rombach et al., 2022), which uses a pre-trained autoencoder to map the high-dimensional pixels to the so-called *latent variables*, which enter into the diffusion model. The latent diffusion model defined on the lower-dimensional latent space has a significantly lower number of parameters to fine-tune than the diffusion model defined on the pixel space.

Rather than fine-tuning the entire LDM, in our method called *differentially private latent diffusion models* (DP-LDMs), we choose to fine-tune only the *attention modules* (and *conditioning embedders* for conditional generation) in the LDM using DP-SGD. As a result, the number of trainable parameters under our approach is only 10% of that of the diffusion models used in DP-Diffusion (Ghalebikesabi et al., 2023) and achieves a better privacy-accuracy trade-off. Our choice – fine-tune the attention modules only – is inspired by recent observations that altering attention modules in large language models (LLMs) substantially alters the models' behaviors (Shi et al., 2023; Hu et al., 2021). Therefore, either manipulating or fine-tuning attention modules can yield a more targeted generation, e.g., targeted for a user-preference (Zhang et al., 2023b) and transferring to a target distribution (You & Zhao, 2023). See Sec. 3 for further discussion.

The combination of considering LDMs and fine-tuning attention modules using DP-SGD is simple, *yet* a solid tool whose potential impact is substantial for the following reasons:

- **Improved performance:** *DP-LDMs outperform* many state-of-the-art methods in FID (Heusel et al., 2017) and downstream classification accuracy when evaluated on several commonly used image benchmark datasets in DP literature. This is due to unique aspects of our proposed method – training DMs in the latent space and fine-tuning only a few selected parameters. This makes our training process considerably more efficient than training a DM from scratch with DP-SGD in DPDM (Dockhorn et al., 2023), or fine-tuning the entire DM with DP-SGD in DP-Diffusion (Ghalebikesabi et al., 2023).

- **Significantly Reduced GPU hours:** Reducing the fine-tuning space in DP-LDMs not only improves the performance but also helps democratize DP image generation using DMs, which otherwise have to rely on massive computational resources only available to a small fraction of the field and would leave a huge carbon footprint. For instance, to generate the CIFAR10 synthetic images using an NVIDIA V100 32GB, a recent work called *DP-API* by Lin et al. (2023) requires 500 GPU hours and DP-Diffusion requires 1250 GPU hours (See Figure 42 in (Lin et al., 2023)). In our case, when using an NVIDIA RTX A4000 16GB GPU (slower than V100 32GB), fine-tuning took 15 GPU hours, and pre-training took 192 GPU hours. Pre-training is not always necessary; we used a single pre-trained LDM for CelebA32, CIFAR10 and Camelyon17.

- **For-the-first-time DP Text-to-Image Generation:** *We push the boundary of what DP-SGD fine-tuned generative models can achieve*, by being the first to produce high-dimensional images (256x256) at a reasonable privacy level. We showcase this in text-conditioned and class-conditioned image generation, where we input a certain text prompt (or a class label) and generate a corresponding image from a DP-fine-tuned LDM for CelebAHQ. These conditional, high-dimensional image generation tasks present more complex and more realistic benchmarks compared to the conventional CIFAR10 and MNIST datasets. These latter datasets, though widely used in DP image generation literature for years, are now rather simplistic and outdated. Our work contributes to narrowing down the large gap between the current state of synthetic image generation in non-DP settings and that in DP settings.

- **Application of DP-LoRA to LDMs:** We apply the *low-rank* approximation (Hu et al., 2021) to the attention modules in DMs to further decrease the number of parameters to fine-tune. Interestingly, the performance from LoRA was slightly worse than that of fine-tuning entire attention modules.

This improvement is due to the large batch size we use, which enhances the signal-to-noise ratio and significantly boosts the performance of the fully fine-tuned model. This phenomenon is similar to what Li et al. (2022) observed when fine-tuning LLMs.

In the following section, we provide relevant background information. We then present our method along with related work and experiments on benchmark datasets.

## 2 Background

We first describe latent diffusion models (LDMs), then the definition of differential privacy (DP) and finally the DP-SGD algorithm, which we will use to train the LDMs in our method.

### 2.1 Latent Diffusion Models (LDMs)

Diffusion Models gradually denoise a normally distributed variable through learning the reverse direction of a Markov Chain of length $T$. Latent diffusion models (LDMs) by Rombach et al. (2022) are a modification of denoising diffusion probabilistic models (DDPMs) by Ho et al. (2020) in the following way. First, Rombach et al. (2022) uses a powerful auto-encoder, consisting of an encoder denoted by Enc and a decoder denoted by Dec . The encoder transforms a high-dimensional pixel representation $\mathbf{x}$ into a lower-dimensional latent representation $\mathbf{z}$ via $\mathbf{z} = \text{Enc}(\mathbf{x})$; and the decoder transforms the lower-dimensional latent representation back to the original space via $\hat{\mathbf{x}} = \text{Dec}(\mathbf{z})$. Rombach et al. (2022) use a combination of a perceptual loss and a patch-based adversarial objective, with extra regularization for better-controlled variance in the learned latent space, to obtain powerful autoencoders (See section 3 in (Rombach et al., 2022) for details). This training loss helps the latent representations to carry equivalent information (e.g., the spatial structure of pixels) as the pixel representations, although the dimensionality of the former is greatly reduced.

Second, equipped with the powerful auto-encoder, Rombach et al. (2022) trains a diffusion model (typically a UNet (Ronneberger et al., 2015)) in the latent space. Training a DM in this space can significantly expedite the training process of diffusion models, e.g., from hundreds of GPU *days* to several GPU *hours* for similar accuracy.

Third, LDMs also contain *attention modules* (Vaswani et al., 2017) that take inputs from a *conditioning embedder*, inserted into the layers of the underlying UNet backbone as the way illustrated in Fig. 1 to achieve flexible conditional image generation (e.g., generating images conditioning on text, image layout, class labels, etc.). The modified UNet is then used as a function approximator $\tau_{\boldsymbol{\theta}}$ to predict an initial noise from the noisy lower-dimensional latent representations at several finite time steps $t$, where in LDMs, the noisy representations (rather than data) follow the diffusion process defined in Ho et al. (2020).

The parameters of the approximator are denoted by $\boldsymbol{\theta} = [\boldsymbol{\theta}^U, \boldsymbol{\theta}^{Attn}, \boldsymbol{\theta}^{Cn}]$, where $\boldsymbol{\theta}^U$ are the parameters of the underlying UNet backbone excluding the parameter of attention modules $\boldsymbol{\theta}^{Attn}$, and $\boldsymbol{\theta}^{Cn}$ are the parameters of the conditioning embedder (We will explain these further in Sec. 3). These parameters are then optimized by minimizing the prediction error defined by

$$\mathcal{L}_{ldm}(\boldsymbol{\theta}) = \mathbb{E}_{(\mathbf{z}_t, y), \tau, t} \left[ \|\tau - \tau_{\boldsymbol{\theta}}(\mathbf{z}_t, t, y)\|_2^2 \right], \tag{1}$$

where $\tau \sim \mathcal{N}(0, I)$, $t$ uniformly sampled from $\{1, \cdots, T\}$, $\mathbf{x}_t$ is the noisy version of the input $\mathbf{x}$ at step $t$, $\mathbf{z}_t = \text{Enc}(\mathbf{x}_t)$ and $y$ is what the model is conditioning on to generate data, e.g., class labels, or a prompt. The function approximator $\tau_{\boldsymbol{\theta}}(\mathbf{z}_t, t, y)$ takes the condition $y$, time step $t$, and latent input $\mathbf{z}_t$ and then maps to an initial noise estimate. Once the approximator is trained, the drawn samples in latent space, $\tilde{\mathbf{z}}$, are transformed back to pixel space through the decoder, i.e., $\tilde{\mathbf{x}} = \text{Dec}(\tilde{\mathbf{z}})$. Our work introduced in Sec. 3 pre-trains both auto-encoder and $\tau_{\boldsymbol{\theta}}$ using public data, then fine-tunes only $\boldsymbol{\theta}_{Attn}, \boldsymbol{\theta}_{Cn}$, the parameters of the attention modules and the conditioning embedded, for private data, using DP-SGD to be explained next.

### 2.2 Differential Privacy (DP)

A mechanism $\mathcal{M}$ is $(\epsilon, \delta)$-DP for a given $\epsilon \geq 0$ and $0 \leq \delta < 1$ if and only if $\Pr[\mathcal{M}(\mathcal{D}) \in S] \leq e^\epsilon \cdot \Pr[\mathcal{M}(\mathcal{D}') \in S] + \delta$ for all possible sets of the mechanism's outputs $S$ and all neighbouring datasets $\mathcal{D}, \mathcal{D}'$ that differ by

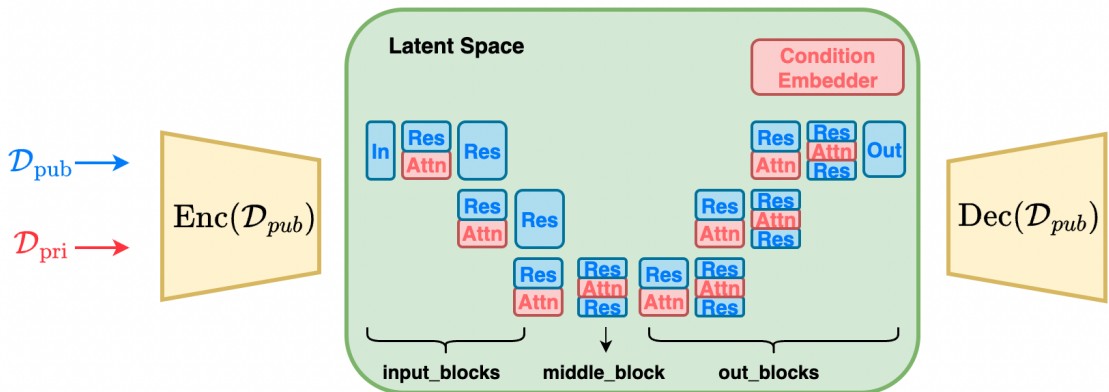

Figure 1: A schematic of DP-LDM. In the non-private step, we pre-train the auto-encoder depicted in yellow (Right and Left) with public data. We then forward pass the public data through the encoder (blue arrow on the left) to obtain latent representations. We then train the diffusion model (depicted in the green box) on the lower-dimensional latent representations. The diffusion model consists of the UNet backbone and added attention modules (in Red) with a conditioning embedder (in Red, at top-right corner). In the private step, we forward pass the private data (red arrow on the left) through the encoder to obtain latent representations of the private data. We then fine-tune only the red blocks, which are attention modules and conditioning embedder, with DP-SGD. Once the training is done, we sample the latent representations from the diffusion model, and pass them through the decoder to obtain the samples in the pixel space.

a single entry. A single entry difference could come from either replacing or including/excluding one entry to/from the dataset $\mathcal{D}$.

One of the most well-known and widely used DP mechanisms is the *Gaussian mechanism*. The Gaussian mechanism adds a calibrated level of noise to a function $\mu : \mathcal{D} \mapsto \mathbb{R}^p$ to ensure that the output of the mechanism is $(\epsilon, \delta)$-DP: $\widetilde{\mu}(\mathcal{D}) = \mu(\mathcal{D}) + n$, where $n \sim \mathcal{N}(0, \sigma^2 \Delta_\mu^2 \mathbf{I}_p)$. Here, $\sigma$ is often called a privacy parameter, which is a function of $\epsilon$ and $\delta$. $\Delta_\mu$ is often called the *global sensitivity* (Dwork et al., 2006; 2014), which is the maximum difference in $L_2$-norm given two neighbouring $\mathcal{D}$ and $\mathcal{D}'$, $||\mu(\mathcal{D}) - \mu(\mathcal{D}')||_2$. Because we are adding noise, the natural consequence is that the released function $\tilde{\mu}(\mathcal{D})$ is less accurate than the non-DP counterpart, $\mu(\mathcal{D})$. This introduces privacy-accuracy trade-offs.

DP-SGD (Abadi et al., 2016) is an instantiation of the Gaussian mechanism in stochastic gradient descent (SGD) by adding an appropriate amount of Gaussian noise to the gradients in every training step, to ensure the parameters of a neural network are differentially private. When using DP-SGD, due to the composability property of DP Dwork et al. (2006; 2014), privacy loss is accumulating over a typically long course of training. Abadi et al. (2016) exploit the subsampled Gaussian mechanism (i.e., applying the Gaussian mechanism on randomly subsampled data) to achieve a tight privacy bound. The *Opacus* package (Yousefpour et al., 2021) implements the privacy analysis in DP-SGD, which we adopt in our method. One thing to note is that we use the **inclusion/exclusion** definition of DP in the experiments as in *Opacus*.

## 3  Differentially private latent diffusion models (DP-LDMs)

In our method, which we call *differentially private latent diffusion models (DP-LDMs)*, we carry out two training steps: non-private and private steps.

**Non-Private Step: Pre-training an autoencoder and a DM using public data.**  Following Rombach et al. (2022), we first pre-train an auto-encoder. The encoder scales down an image $\mathbf{x} \in \mathbb{R}^{H \times W \times 3}$ to a 3-dimensional latent representation $\mathbf{z} \in \mathbb{R}^{h \times w \times c}$ by a factor of $f$, where $f = H/h = W/w$. This 3-dimensional latent representation is chosen to take advantage of image-specific inductive biases that the UNet contains, e.g., 2D convolutional layers. Following Rombach et al. (2022), we also train the autoencoder by minimizing a

combination of different losses, such as perceptual loss and adversarial loss, with some form of regularization. See Appendix Sec. A.1 for details. As noted by Rombach et al. (2022), we also observe that a mild form of downsampling performs the best, achieving a good balance between training efficiency and perceptually decent results. See Appendix Sec. A.1 for details on different scaling factors $f = 2^m$, with a different value of $m$. Training an auto-encoder does not incur any privacy loss, as we use public data $\mathcal{D}_{pub}$ considered to be similar to private data $\mathcal{D}_{priv}$ at hand. The trained autoencoder is, therefore, a function of public data: an encoder $\text{Enc}(\mathcal{D}_{pub})$ and a decoder $\text{Dec}(\mathcal{D}_{pub})$.

A forward pass through the trained encoder $\text{Enc}(\mathcal{D}_{pub})$ gives us a latent representation of each image, which we use to train a diffusion model. As in (Rombach et al., 2022), we consider a modified UNet for the function approximator $\tau_{\boldsymbol{\theta}}$ shown in Fig. 1. We minimize the loss given in eq. 1 to estimate the parameters of $\tau_{\boldsymbol{\theta}}$ as:

$$\boldsymbol{\theta}^{U}_{\mathcal{D}_{pub}}, \boldsymbol{\theta}^{Attn}_{\mathcal{D}_{pub}}, \boldsymbol{\theta}^{Cn}_{\mathcal{D}_{pub}} = \arg\min_{\boldsymbol{\theta}} \ \mathcal{L}_{ldm}(\boldsymbol{\theta}). \tag{2}$$

Since we use public data, there is no privacy loss incurred in estimating the parameters, which are a function of public data $\mathcal{D}_{pub}$.

**Private Step: Fine-tuning attention modules & conditioning embedder for private data.** Given a pre-trained diffusion model, we fine-tune the attention modules and a conditioning embedder using our private data.

For the models with the conditioned generation, the attention modules refer to the spatial transformer blocks shown in Fig. 2(a) which contains cross-attentions and multiple heads. For the models with an unconditional generation, the attention modules refer to the attention blocks shown in Fig. 2(b). Consequently, the parameters of the attention modules, denoted by $\boldsymbol{\theta}^{Attn}$, differ, depending on the conditioned or unconditioned cases. The conditioning embedder only exists in the conditioned case. Depending on the different modalities the model is trained on, the conditioning embedder takes a different form. For instance, if the model generates images conditioning on the class labels, the conditioning embedder is simply a class embedder, which embeds class labels to a latent dimension. If the model conditions on language prompts, the embedder can be a transformer.

The core part of the spatial transformer block and the attention block is the attention layer, which has the following parameterization. Here, we explain it under the conditioned case:

$$\text{Attention}(\psi_i(\mathbf{z}_t), \phi(y); Q, K, V) = \text{softmax}\left(\frac{QK^T}{\sqrt{d_k}}\right) V \ \in \mathbb{R}^{N \times d_k}, \tag{3}$$

where $\psi_i(\mathbf{z}_t) \in \mathbb{R}^{N \times d^i}$ is an intermediate representation of the latent representation $\mathbf{z}_t$ through the $i$th residual convolutional block in the backbone UNet, and $\phi(y) \in \mathbb{R}^{M \times d_c}$ is the embedding of what the generation is conditioned on (e.g., class labels, or CLIP embedding). Furthermore,

$$Q = \psi_i(\mathbf{z}_t)W_Q^{(i)\top}, \ K = \phi(y)W_K^{(i)\top}, \ V = \phi(y)W_V^{(i)\top}, \tag{4}$$

where the parameters are denoted by $W_Q^{(i)} \in \mathbb{R}^{d_k \times d^i}$; $W_K^{(i)} \in \mathbb{R}^{d_k \times d_c}$; and $W_V^{(i)} \in \mathbb{R}^{d_k \times d_c}$. Unlike the conditioned case, where the key ($K$) and value ($V$) vectors are computed as a projection of the conditioning embedder, the key and value vectors are a projection of the pixel embedding $\psi_i(\mathbf{z}_t)$ only in the case of the unconditioned model. We run DP-SGD to fine-tune these parameters to obtain differentially private $\boldsymbol{\theta}^{Attn}_{\mathcal{D}_{priv}}$ and $\boldsymbol{\theta}^{Cn}_{\mathcal{D}_{priv}}$, starting from $\boldsymbol{\theta}^{Attn}_{\mathcal{D}_{pub}}, \boldsymbol{\theta}^{Cn}_{\mathcal{D}_{pub}}$. Our algorithm is given in Algorithm 1.

**Why fine-tune attention modules only?** The output of the attention in eq. 3 assigns a high focus to the features that are more important, by zooming into what truly matters in an image depending on a particular context, e.g., relevant to what the image is conditioned on. This can be quite different when we move from one distribution to the other. By fine-tuning the attention modules (together with the conditioning embedder when conditioned case), we effectively transfer what we learned from the public data distribution to the private data distribution. However, if we fine-tune other parts of the model, e.g., the ResBlocks, the fine-tuning of these blocks can make a large change in the features themselves, degrading the performance. See

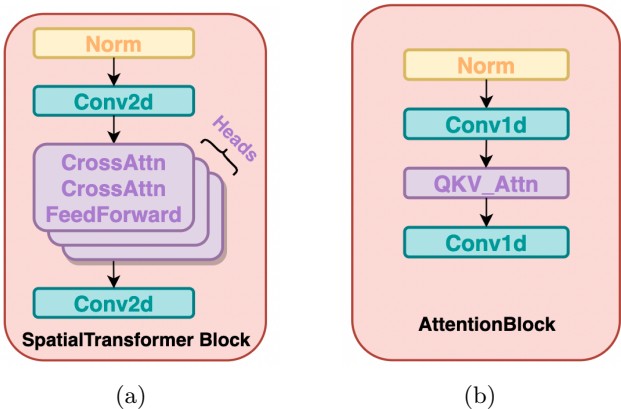

Figure 2: (a) SpatialTransformer Block; (b) AttentionBlock

---

**Algorithm 1** DP-LDM

---

**Input:** Latent representations and conditions if conditional: $\{(\mathbf{z}_i, y_i)\}_{i=1}^{N}$, a pre-trained model $\boldsymbol{\theta}$, number of iterations $P$, mini-batch size $B$, clipping norm $C$, learning rate $\eta$, privacy parameter $\sigma$ corresponding to $(\epsilon, \delta)$-DP. Denote $\hat{\boldsymbol{\theta}} = \{\boldsymbol{\theta}^{Attn}, \boldsymbol{\theta}^{Cn}\}$

**for** $p = 1$ **to** $P$ **do**

  **Step 1**. Take a mini-batch $B_p$ uniformly at random with a sampling probability, $q = B/N$

  **Step 2**. For each sample $i \in B_p$ compute the gradient:

  $g_p(\mathbf{z}_i, y_i) = \nabla_{\hat{\boldsymbol{\theta}}_p} \mathcal{L}_{ldm}(\hat{\boldsymbol{\theta}}_p, \mathbf{z}_i, y_i)$

  **Step 3**. Clip the gradient:

  $\hat{g}_p(\mathbf{z}_i, y_i) = g_p(\mathbf{z}_i, y_i)/\max(1, \|g_p(\mathbf{z}_i, y_i)\|_2/C)$

  **Step 4**. Add noise:

  $\tilde{g}_p = \frac{1}{B} \left( \sum_{i=1}^{B} \hat{g}_p(\mathbf{z}_i, y_i) + \mathcal{N}(0, \sigma^2 C^2 I) \right)$

  **Step 5**. Gradient descent: $\hat{\boldsymbol{\theta}}_{p+1} = \hat{\boldsymbol{\theta}}_p - \eta \tilde{g}_p$

**end for**

**Return:** $(\epsilon, \delta)$-differentially private $\hat{\boldsymbol{\theta}}_P = \{\boldsymbol{\theta}_P^{Attn}, \boldsymbol{\theta}_P^{Cn}\}$

---

Sec. 5. The idea of fine-tuning attention blocks is explored elsewhere. In fine-tuning large language models, existing work introduces a few new parameters to transformer attention blocks, and those new parameters are fine-tuned (Yu et al., 2022; Hu et al., 2021) to adapt to new distributions. In the context of pre-trained diffusion models, adding, modifying, and controlling attention layers are gaining popularity for tasks such as image editing and text-to-image generation (Hertz et al., 2022; Park et al., 2023; Zhang et al., 2023a; You & Zhao, 2023).

**Which public dataset to use given a private dataset?** This is an open question in transfer learning literature. Generally, if the two datasets are close to each other in *some* sense, they are assumed to be a better pair. However, similarity *in what sense* has to be chosen differently depending on a particular data domain and appropriately privatized if private data is used in this step. For instance, if the data is in 2D space, similarity in L2-distance sense might make sense. Our data lies in a high dimensional pixel space, where judging similarity in the FID sense is widely used (when comparing two image distributions using each distribution's image samples). Hence, we use FID as a proxy to judge the similarity between two *image* datasets (See Sec. 5.1 how we privatize this step). In other datasets, out of the image domain, there could be a more appropriate metric to use than FID, e.g., in the case of discrete data, kernel-based distance metrics with an appropriately chosen kernel could be more useful.

## 4 Related Work

Initial efforts on generating high-dimensional image data with differential privacy have focused on leveraging advanced generative models to achieve better differentially private synthetic data (Hu et al., 2023). Some of them (Xie et al., 2018; Torkzadehmahani et al., 2019; Frigerio et al., 2019; Yoon et al., 2019; Chen et al., 2020) utilize generative adversarial networks (GANS) (Goodfellow et al., 2014), or trained GANs with the PATE structure (Papernot et al., 2017). Other works have employed variational autoencoders (VAEs) (Acs et al., 2018; Jiang et al., 2022; Pfitzner & Arnrich, 2022), or proposed customized structures (Harder et al., 2021; Vinaroz et al., 2022; Cao et al., 2021; Liew et al., 2022a; Harder et al., 2023). For instance, Harder et al. (2023) pretrained perceptual features using public data and privatized only data-dependent terms using maximum mean discrepancy.

Limited works have so far delved into privatizing diffusion models. Dockhorn et al. (2023) develop a DP score-based generative models (Song et al., 2021) using DP-SGD, applied to relatively simple datasets such as MNIST, FashionMNIST and CelebA (downsampled to 32×32). Ghalebikesabi et al. (2023) fine-tune the ImageNet pre-trained diffusion model (DDPM) (Ho et al., 2020) with more than 80 M parameters using DP-SGD for CIFAR-10. We instead adopt a different model (LDM) and fine-tune only the small part of the DM in our model to achieve better privacy-accuracy trade-offs. As concurrent work to ours, Lin et al. (2023) propose a DP-histogram mechanism to generate synthetic data through the utilization of publicly accessible APIs.Another concurrent work to ours, DP-Promise (Wang et al., 2024) leverages the diffusion model's noise during the forward process to improve the privacy-accuracy trade-offs in diffusion model training.

## 5 Experiments

We demonstrate the performance of our method in comparison with the state-of-the-art methods in DP data generation, using several combinations of public/private data of different levels of complexity at varying privacy levels.

**Implementation.** We implemented DP-LDMs in PyTorch Lightning (Paszke et al., 2019) building on the LDM codebase by Rombach et al. (2022) and Opacus (Yousefpour et al., 2021) for DP-SGD training. Several recent papers present the importance of using large batches in DP-SGD training to improve accuracy at a fixed privacy level (Ponomareva et al., 2023; De et al., 2022; Bu et al., 2022). To incorporate this finding in our work, we wrote custom batch splitting code that integrates with Opacus and Pytorch Lightning, allowing us to test arbitrary batch sizes. Our DP-LDM also improves significantly with large batches as will be shown soon, consistent with the findings in recent work. For our experiments incorporating LoRA, we use the loralib (Hu et al., 2021) Python library.

**Datasets**[1] **and Evaluation.** We list all the private and public dataset pairs with corresponding evaluation metrics in Table 1. Regarding high-quality generation, we use CelebAHQ for class conditional generation and Multi-Modal-CelebAHQ (MM-CelebAHQ) for text-to-image generation. Our choice of evaluation metric is either based on standard practice or following previous work to do a fair comparison. We measure the model performance by computing the Fréchet Inception Distance (FID) (Heusel et al., 2017) between the generated samples and the real data. For downstream task, we consider CNN (LeCun et al., 2015), ResNet-9 (He et al., 2016), and WRN40-4 (Zagoruyko & Komodakis, 2017) to evaluate the classification performance of synthetic data. Each number in our tables represents an average value across three independent runs, with a standard deviation (unless stated otherwise). Note that some standard deviation values are reported as 0.0 due to rounding. Values for comparison methods are taken from their papers, with an exception for the DP-MEPF comparison to CelebAHQ, which we ran their code by loading this data.

### 5.1 Comparisons to State-of-the-art methods

We start with testing our method on private and public dataset pairs at varying complexity, which are generally considered to be relatively similar to each other. In particular, we present the results of transferring from

---

[1]Dataset licenses: MNIST: CC BY-SA 3.0; CelebA: see https://mmlab.ie.cuhk.edu.hk/projects/CelebA.html; CIFAR-10: MIT; Camelyon17: CC0

| Private Dataset | Public Dataset | Similarity Evaluation | Downstream Classifiers |
|---|---|---|---|
| MNIST (LeCun & Cortes, 2010) | EMNIST(letter) (Cohen et al., 2017) | - | CNN, WRN40-4 |
| CIFAR-10 (Krizhevsky et al., 2009) | ImageNet32 (Deng et al., 2009) | FID | ResNet-9, WRN40-4 |
| Camelyon17-WILDS (Koh et al., 2021) | ImageNet32 (Deng et al., 2009) | - | WRN40-4 |
| CelebA32 (Liu et al., 2015) | ImageNet32 (Deng et al., 2009) | FID | - |
| CelebA64 (Liu et al., 2015) | ImageNet64 (Deng et al., 2009) | FID | ResNet-9 |
| CelebAHQ (Karras et al., 2018) | ImageNet256 (Deng et al., 2009) | FID | - |
| MM-CelebAHQ (Xia et al., 2021) | LAION-400M (Schuhmann et al., 2021) | FID | - |

Table 1: Private and public dataset pairs, with corresponding evaluation metric and choices of classifiers.

| | | $\epsilon = 10$ | $\epsilon = 5$ | $\epsilon = 1$ | $\epsilon = 0.67$ |
|---|---|---|---|---|---|
| CIFAR-10 32x32 ($\delta = 10^{-5}$) | **DP-LDM(ours)** | **8.4 ± 0.2** | **13.4 ± 0.4** | **22.9 ± 0.5** | |
| | DP-Diffusion | 9.8 | 15.1 | 25.2 | |
| | DP-MEPF ($\phi_1, \phi_2$) | 29.1 | 30.0 | 54.0 | |
| | DP-MEPF ($\phi_1$) | 30.3 | 35.6 | 56.5 | |
| | DPDM | 97.7 | - | - | |
| | DP-API | - | - | - | **7.87** |
| | DP-Promise | 17.9 | 18.9 | 21.8 | |
| CelebA 32x32 ($\delta = 10^{-6}$) | **DP-LDM(ours)** | 16.2 ± 0.2 | 16.8 ± 0.3 | 25.8 ± 0.9 | |
| | DP-MEPF ($\phi_1$) | 16.3 | 17.2 | 17.2 | |
| | DP-GAN (pre-trained) | 58.1 | 66.9 | 81.3 | |
| | DPDM | 21.2 | - | 71.8 | |
| | DP Sinkhorn | 189.5 | - | - | |
| | DP-Promise | **6.0** | **6.5** | **9.0** | |
| CelebA 64x64 ($\delta = 10^{-6}$) | **DP-LDM(ours)** | **14.3 ± 0.1** | **16.1 ± 0.2** | 21.1 ± 0.2 | |
| | DP-MEPF ($\phi_1$) | 17.4 | 16.5 | **20.4** | |
| | DP-GAN (pre-trained) | 57.1 | 62.3 | 72.5 | |
| | DPDM | 78.3 | - | - | |
| | DP-Promise | 25.3 | 26.2 | 29.1 | |

Table 2: FID scores (lower is better) for synthetic CIFAR-10, CelebA32, and CelebA64 data, in comparison with DP-Diffusion (Ghalebikesabi et al., 2023), DP-MEPF (Harder et al., 2023), DPDM (Dockhorn et al., 2023), DP-GAN (Xie et al., 2018), DP Sinkhorn (Cao et al., 2021), and DP-Promise (Wang et al., 2024)

Imagenet to CIFAR-10 and CelebA distributions and from EMNIST to MNIST distribution. Additionally, to test our method's effectiveness at transferring knowledge across a large domain gap, we present the results of transferring from Imagenet to Camelyon17-WILDS. Further details on these experiments are available in appendix A.

**FID.** Comparison to other SOTA methods in terms of FID (the lower the better) is illustrated in Table 2. When tested on CIFAR-10, our DP-LDM outperforms other methods at all epsilon levels ($\epsilon = 1, 5, 10$ and $\delta = 10^{-5}$) except for DP-API. Our FID values correspond to the case where only 9-16 attention modules are fine-tuned (i.e., fine-tuning only 10% of trainable parameters in the model) and the rest remain fixed. See Table 13 for ablation experiments for fine-tuning different attention modules. When tested on CelebA32, DP-Promise achieved the best FID at all $\epsilon$ levels by a large margin compared to other existing methods. When

tested on CelebA64, our unconditional LDM achieves new SOTA results at $\epsilon = 10, 5$ and are comparable to DP-MEPF at $\epsilon = 1$. Samples are available in Fig. 11.

| | | $\epsilon = 10$ | $\epsilon = 5$ | $\epsilon = 1$ |
|---|---|---|---|---|
| CIFAR-10 ResNet-9 | **DP-LDM(ours)** | **65.3 ± 0.3** | **59.1 ± 0.2** | **51.3 ± 0.1** |
| | DP-MEPF $(\phi_1, \phi_2)$ | 48.9 | 47.9 | 28.9 |
| | DP-MEPF $(\phi_1)$ | 51.0 | 48.5 | 29.4 |
| | DP-MERF | 13.2 | 13.4 | 13.8 |
| CIFAR-10 WRN-40-4 | **DP-LDM(ours)** | **78.6 ± 0.3** | - | - |
| | DP-Diffusion | 75.6 | - | - |
| CelebA64 ResNet-9 | **DP-LDM(ours)** | **96.4 ± 0.0** | **96.0 ± 0.0** | **94.5 ± 0.0** |
| | DP-MEPF $(\phi_1)$ | 93.9 ± 2.1 | 93.7 | 82.9 |
| MNIST CNN | **DP-LDM(ours)** | 97.4 ± 0.1 | **96.8** | **95.9 ± 0.1** |
| | DPDM | **98.1** | - | 95.2 |
| MNIST WRN-40-4 | **DP-LDM(ours)** | 97.5 ± 0.0 | - | - |
| | DP-Diffusion | **98.6** | - | - |
| Camelyon17-WILDS WRN-40-4 | **DP-LDM(ours)** | 85.4 ± 0.0 | - | - |
| | DP-Diffusion | **91.1** | - | - |
| | DP-API | 80.5 | | |

Table 3: Classification accuracies (higher is better) evaluated on real test data, when the classifiers are trained with synthetic CIFAR-10, CelebA64, and MNIST datasets. Comparison methods include DP-MEPF (Harder et al., 2023), DP-MERF (Harder et al., 2021), DP-Diffusion (Ghalebikesabi et al., 2023), and DPDM (Dockhorn et al., 2023). For Camelyon17 dataset, following Ghalebikesabi et al. (2023) and Lin et al. (2023), we set $\delta = 3 \cdot 10^{-6}$. Each classifier's architecture is written below each name of data. We choose to use these classifiers by following previous works (Harder et al., 2023; Ghalebikesabi et al., 2023; Dockhorn et al., 2023).

**Downstream Classification.** FID can be viewed as a fidelity metric, serving as a proxy for the utility of the synthetic data. To directly present the utility results of the model, we also consider accuracy on the classification task, which is listed in Table 3. All the classifiers are trained with 50K synthetic samples and then evaluated on real data samples. For each dataset, we follow previous work to choose classifier models for a fair comparison.

When tested on CIFAR-10, DP-LDM again outperforms others except DP-API. Compared to DP-LDM, the performance of DP-API seems more susceptible to the amount of domain shift from public to private data distributions. When there is a small domain gap, e.g., from Imagenet (public data) to CIFAR10 (private data), DP-API performs better than DP-LDM. However, when there is a large domain shift between public data (Imagenet) to private data (Camelyon17), the accuracies of downstream classifier (WRN-40-4) trained with each synthetic dataset are 85.7 for DP-LDMs and 80.5 for DP-API, respectively, at $\epsilon = 10$. Hence, DP-LDMs seem better suited than DP-API when the domain shift is large as fine-tuning a small part of diffusion models in DP-LDMs helps incorporate such a shift effectively.

For testing on CelebA64, we began with an LDM pre-trained on conditional ImageNet at the same resolution, and then fine-tuned it on CelebA where the (binary) class labels were given by the "Male" attribute. Our method achieves a new SOTA performance at all epsilon levels.

When tested on MNIST, we surpass the previous methods at $\epsilon = 1$ and achieve comparable results at $\epsilon = 10$. One thing to note is that DPDM takes 1.75M parameters and 192 GPU hours to achieve 98.1 accuracy, and DP-Diffusion takes 4.2M parameters (GPU hours not showing), while our methods takes only 0.8M parameters and 10 GPU hours to achieve 97.4 accuracy, which significantly reduces the parameters and saves much computing resources.

When tested on Camelyon17, our method seems to underperform DP-Diffusion (Ghalebikesabi et al., 2023). However, this could be due to their use of a pre-trained WRN-40-4 classifier (pre-trained with ImageNet32). Ghalebikesabi et al. (2023) has not published their code and the pre-trained classifier (using ImageNet32) that they started fine-tuning with, so we could not directly compare the results between ours and theirs.

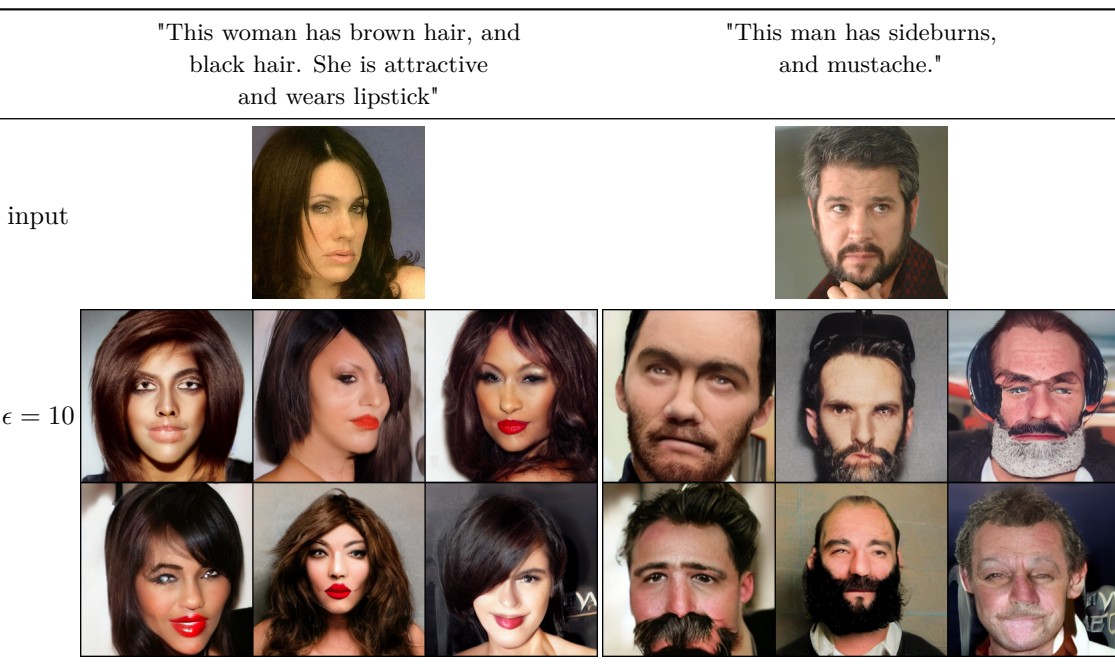

Figure 3: Text-to-image generation of $256 \times 256$ CelebAHQ with prompts at $\epsilon = 10$. FID: 15.6

**Why do we select EMNIST as a public dataset?** Previous work (Harder et al., 2023) used SVHN as a public dataset to MNIST since they are both number images. However, SVHN and MNIST differ significantly (SVHN contains several digits per image with 3 channels while MNIST contains one digit per image with 1 channel). So we considered other, more similar datasets such as EMNIST and KMNIST as public dataset candidates. We used FID scores to judge the closeness between public and private data, using privatized FID statistics by following the mechanisms used in (Park et al., 2017). See Table 11 and Appendix Sec. A.2 which verifies our choice of EMNIST.

## 5.2 Differentially private high-quality image generation

With the latent representations of the inputs, LDMs can better improve DP training. To the best of our knowledge, we are the first to achieve high-dimensional differentially private generation.

**Text-to-image generation.** For text-to-image generation, we fine-tune the LDM models pretrained with LAION-400M (Schuhmann et al., 2021) for MM-CelebAHQ ($256 \times 256$). Each image is described by a caption, which is fed to the conditioning embedder, *BERT* (Devlin et al., 2018). We freeze the BERT embedder as well during fine-tuning attention modules to reduce the trainable parameters, then we bring back BERT for sampling. DP-LDM achieves FID scores of 15.6 for $\epsilon = 10$. As illustrated in Fig. 3, the samples are faithful to our input prompts, but not identical to the training sample, unlike the memorization behavior of the non-private Stable Diffusion (Carlini et al., 2023).

**Class conditional generation.** We build our model on the LDM model provided by Rombach et al. (2022) which is pretrained on Imagenet at a resolution of $256 \times 256$. Following our experiments in Sec. 5.1, we fine-tune all of the SpatialTransformer blocks. While CelebAHQ does not provide class labels, each image is associated with 40 binary attributes. We choose the attribute "Male" to act as a binary class label for each image. Generated samples are available in Fig. 4 along with FID values. Compared to DP-MEPF, based on the FID scores and perceptual image quality, DP-LDM is better suited for generating detailed, plausible samples from the high-resolution dataset at a wide range of privacy levels.

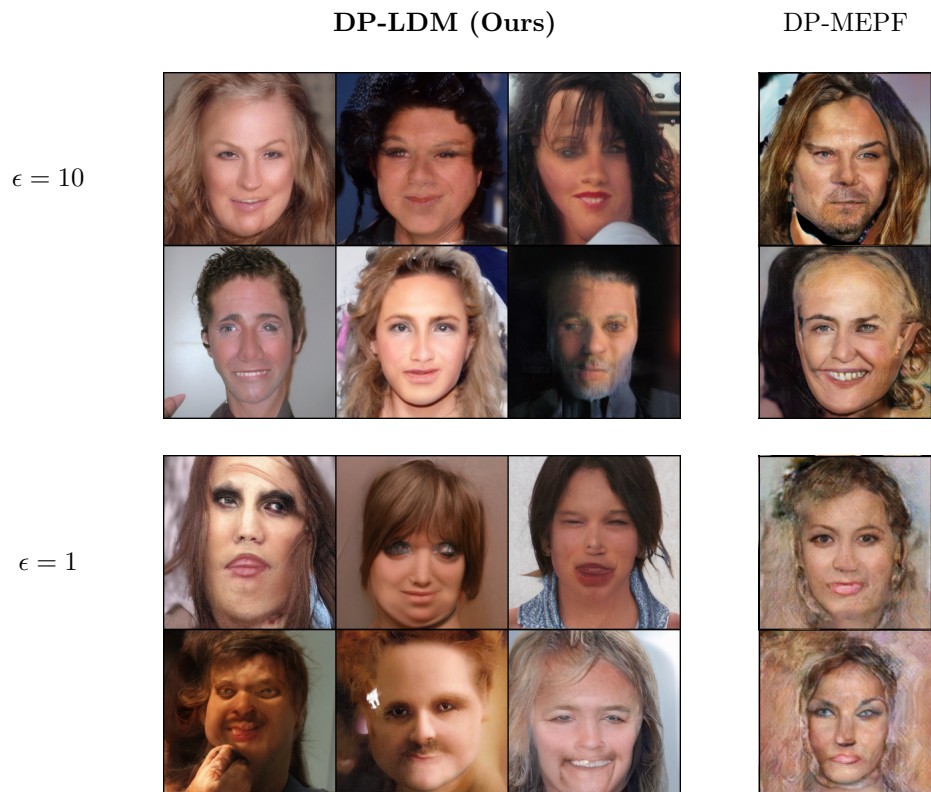

Figure 4: Synthetic $256 \times 256$ CelebA samples generated at varying $\epsilon$. Samples for DP-MEPF are generated from code available in Harder et al. (2023). We computed FID between our generated samples and the real data and achieve FIDs of $19.0 \pm 0.0$ at $\epsilon = 10$, $20.5 \pm 0.1$ at $\epsilon = 5$, and $25.6 \pm 0.1$ at $\epsilon = 1$. DP-MEPF achieves an FID of 41.8 at $\epsilon = 10$ and 101.5 at $\epsilon = 1$.

|  |  | $\epsilon = 10$ | $\epsilon = 5$ | $\epsilon = 1$ |
|---|---|---|---|---|
| Batch | **8192** | **14.3 ± 0.1** | **16.1 ± 0.2** | **21.1 ± 0.2** |
| Size | 2048 | 16.2 ± 0.2 | 17.1 ± 0.2 | 22.1 ± 0.1 |
|  | 512 | 17.2 ± 0.1 | 18.0 ± 0.1 | 22.3 ± 0.2 |
| Fine-tune | **Attention Modules** | **8.4 ± 0.2** | **13.4 ± 0.4** | **22.9 ± 0.5** |
| Different | Resblocks | 105.5 | 127.7 | 149.7 |
| Parts | Out_blocks | 45.8 | 48.3 | 57.2 |
|  | Input_blocks | 54.2 | 56.9 | 70.4 |

Table 4: **Top**. Effect of increasing batch size on FID (CelebA64). At a fixed epsilon level, larger batches improve FIDs. **Bottom**. Effect of fine-tuning only the selected part of the model (CIFAR-10). At a fixed epsilon level, fine-tuning attention modules only achieves best results.

## 5.3 Ablation studies

In this section, we present ablation studies for strategically improving performances and reducing parameters.

**Increasing batch size.** Table 4 (Top) shows results for DP-LDM trained with CelebA64 under different training batch sizes, which provides evidence suggesting that training with larger batch sizes improves the performance of the model.

**Fine-tuning attention modules at different layers.** To further reduce more trainable parameters, we consider fine-tuning only a portion of attention modules See Table 13 for CIFAR-10 and Table 12 for

| | **DP-LDM** | LoRA (differ in rank) | | | | |
|---|---|---|---|---|---|---|
| Rank | | 64 | 8 | 4 | 2 | 1 |
| $\epsilon = 10$ | **14.3** | 16.0 | 17.0 | 18.5 | 20.6 | 22.6 |
| $\epsilon = 5$ | **16.1** | 18.2 | 17.6 | 19.4 | 21.3 | 22.7 |
| $\epsilon = 1$ | **21.1** | 22.3 | 23.1 | 21.5 | 24.3 | 26.3 |
| # Parameters | 8.0M | 1.3M | 16k | 80k | 40k | 20k |
| trainable/total | 11.03% | 1.74% | 0.22% | 0.11% | 0.06% | 0.03% |

Table 5: FID scores (lower is better) for incorporating LoRA into DP-LDM with different ranks with CelebA64. Each model was trained for 70 epochs.

| Camelyon17 | | | CIFAR10 | |
|---|---|---|---|---|
| methods | FID | Classification Acc (WRN-40-4) | methods | FID |
| DP-LDM (layer=13-16) | **64.86** | **85.4** | DP-LDM (layer=9-16) | **8.4** |
| DP-LDM (layer=1-16) | 69.55 | - | DP-LDM (layer=1-16) | 25.8 |
| DP-LoRA (rank=8) | 66.41 | 82.6 | DP-LoRA (rank=4) | 26.77 |

Table 6: FID scores and Classification Accuracy of DP-LoRA trained for Camelyon17 and CIFAR10, at $\epsilon = 10$. The best DP-LoRA models (the best ranks shown in parentheses) still lag the best DP-LDMs (the layer ablations shown in parentheses).

MNIST. See also Appendix Sec. A.3 and Appendix Sec. A.2.3 for details. The best results are achieved when fine-tuning the attention modules in the out_blocks in the UNet (out_blocks shown in Fig. 1), consistently throughout all datasets we tested. If a limited privacy budget is given, we suggest fine-tuning the attention modules in the out_blocks only to achieve better accuracies.

**Fine-tuning a different part of the Unet.** Previous results focused on fine-tuning attention modules at varying layers. Here, we present the performance of fine-tuning a different part of the Unet while the rest of the model is frozen. In Table 4 (Bottom), we show the FID scores evaluated on synthetic CIFAR10 images. The main takeaway messages are (a) fine-tuning Resblocks hurts the performance, possibly because the learned features during the pre-training stage are altered, and (b) fine-tuning out_blocks is more useful than input_blocks, while the best strategy is fine-tuning the attention modules in the out_blocks.

## 5.4 DP-LoRA: Applying LoRA to LDM and fine-tuning with attention modules using DP-SGD.

Previous work (Yu et al., 2022) has explored LoRA during training to reduce the fine-tuning parameters. We performed additional experiments by applying LoRA to the QKV matrices in all the attention modules for CelebA64, CIFAR10, and Camelyon17 datasets. In LoRA, each QKV matrix is reparameterized as $W_{\text{new}} = W_{\text{pretrained}} + \frac{\alpha}{\text{rank}} \cdot BA$, where $W_{\text{new}}$ is the desired result of finetuning, $W_{\text{pretrained}}$ is the pretrained weight matrix, $A$ and $B$ are the low rank matrices, and $\alpha$ is a scaling hyperparameter. Following Hu et al. (2022), we set $\alpha$ to be equal to the first rank we try for each experiment, which in all of our runs is 1. In DP-LoRA, A and B are updated during fine-tuning with DP-SGD and $W_{\text{pretrained}}$ is frozen. As shown in Table 5 and Table 6, DP-LoRA was not particularly useful under the LDMs and our current DP-LDM still outperformed. A possible explanation could be the phenomenon previously observed in fine-tuning LLMs (Li et al., 2022): They reasoned that the large batch size improves the signal-to-noise ratio (shown in Fig 3 [1] of Li et al. (2022)), which significantly helps improve the performance of the model with full updates, compared to low-rank updates.

For more curious readers, we show the empirical distances between $W_{\text{new}}$ and $W_{\text{pretrained}}$ for the LoRA models fine-tuned for CelebA64. We compute the $\ell_2$ distance between the prerained and finetuned *qkv.weight* matrices for DP-LDM (vanilla), and the $F$-norm of $\frac{\alpha}{rank} \cdot BA$ for DP-LoRA, in Table 7. Two remarks: First, given that the domain shift between ImageNet and CelebA is relatively large, the adaptation matrices in LoRA, in some cases (e.g., at $\epsilon = 10$), completely overwhelm $W_{\text{pretrained}}$. We suspect that the larger F. norms in the case of DP-LoRA (relative to DP-LDM) are due to its updates being restricted to be low-rank.

| | $\epsilon$ | Layer number | | | | | | | | | | | | | | | | |
|---|---|---|---|---|---|---|---|---|---|---|---|---|---|---|---|---|---|---|
| | | 1 | 2 | 3 | 4 | 5 | 6 | 7 | 8 | 9 | 10 | 11 | 12 | 13 | 14 | 15 | 16 | All Params |
| pretrained | | 18.85 | 17.81 | 27.98 | 27.82 | 44.53 | 45.51 | 46.46 | 46.01 | 46.39 | 46.57 | 28.68 | 28.93 | 30.3 | 20.81 | 20.23 | 18.03 | 136.47 |
| finetuned | 10 | 6.36 | 6.60 | 13.43 | 13.47 | 26.94 | 26.95 | 26.95 | 26.92 | 26.95 | 27.02 | 13.75 | 13.84 | 13.73 | 6.83 | 6.49 | 6.00 | 74.15 |
| DP-LDM | 5 | 6.52 | 6.68 | 13.43 | 13.46 | 26.93 | 26.94 | 26.94 | 26.91 | 26.93 | 26.99 | 13.62 | 13.67 | 13.59 | 6.76 | 6.58 | 6.33 | 74.09 |
| | 1 | 6.66 | 6.73 | 13.43 | 13.44 | 26.92 | 26.92 | 26.93 | 26.88 | 26.91 | 26.95 | 13.48 | 13.49 | 13.43 | 6.65 | 6.66 | 6.65 | 73.99 |
| finetuned | 10 | 17.09 | 19.68 | 53.05 | 50.69 | 141.35 | 145.08 | 145.71 | 143.37 | 141.03 | 133.56 | 47.92 | 41.20 | 41.25 | 15.41 | 14.48 | 14.49 | 364.64 |
| DP-LoRA | 5 | 6.49 | 6.90 | 13.54 | 14.10 | 28.55 | 29.08 | 28.42 | 30.26 | 29.01 | 28.75 | 15.72 | 14.90 | 16.02 | 7.89 | 6.35 | 5.68 | 79.90 |
| | 1 | 7.14 | 8.30 | 17.34 | 16.98 | 37.38 | 37.50 | 37.41 | 38.14 | 36.50 | 36.90 | 15.32 | 14.79 | 15.93 | 7.34 | 6.94 | 7.32 | 99.61 |

Table 7: $F$-norms of pretrained parameters (QKV matrices) are in the top part of the table; the $\ell_2$ distances between the pretrained and finetuned QKV.weight matrices for DP-LDM (Middle); DP-LoRA (Bottom). For each layer, the $F$-norm of $\frac{\alpha}{rank} \cdot BA$ is computed, and for the final column, the $\frac{\alpha}{rank} \cdot BA$ matrices are first concatenated across all layers, then the norms are computed. Following the original code of LoRA, we set $\alpha = 1$.

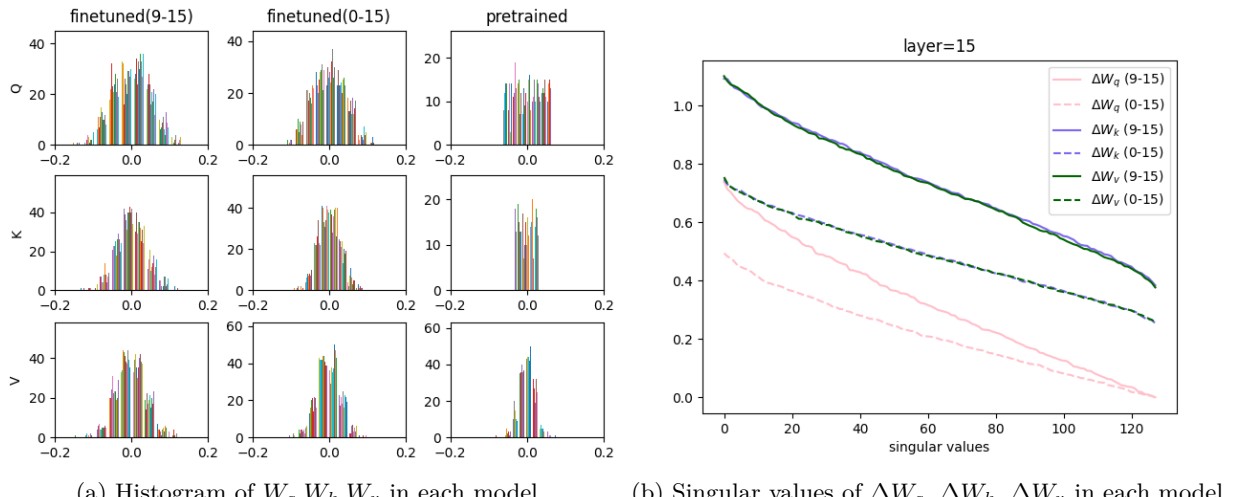

(a) Histogram of $W_q, W_k, W_v$ in each model.    (b) Singular values of $\Delta W_q$, $\Delta W_k$, $\Delta W_v$ in each model.

Figure 5: The Q,K,V matrices in cross-attention modules in two fine-tuned models for CIFAR10 at $\epsilon = 10$. Fine-tuned(0-15) indicates attention modules of all layers are fine-tuned, while Fine-tuned(9-15) indicates only the attention modules at layers 9-15 are fine-tuned. (a) Each row corresponds to $W_q$(top), $W_k$(middle), and $W_v$(bottom). (b) Pink represents $\Delta W_q$, blue $\Delta W_k$, and green $\Delta W_v$, respectively. Solid lines represent the model(9-15) and dotted lines represent the model(0-15).

While DP-LDM may find a configuration of parameter values that achieve low loss (and good performance) relatively closer to the initial parameters, DP-LoRA may need to search further in parameter space for a set of parameter values that achieves similar performance. Second, the fact that DP-LoRA gets the best FID score at rank 64 at $\epsilon = 10$ (the smallest amount of noise we tried out) while it does at rank 4 at $\epsilon = 1$ and rank 8 at $\epsilon = 5$, the optimal adaptation matrix in the UNet might not be necessarily rank-deficient as in the transformer case. Next, we visualize the fine-tuned attention modules in an attempt to gain insights into our fine-tuned models.

### 5.5 Visualization of fine-tuned attention modules

We first show the histogram of $W_q, W_k, W_v$ matrices in cross-attention modules before and after fine-tuning for CIFAR10 in (a) of Fig. 5. Because we fine-tune only the attention modules while the intermediate representations are fixed, the distributions over $W_q, W_k, W_v$ matrices after fine-tuning (first and second columns) are significantly different from those before fine-tuning (third column). When fine-tuning a smaller number of layers (9-15), larger changes have to be made per layer (indicated by the large spread in the histogram for $W_q, W_k, W_v$ matrices) to compensate for the distributional shift from $\mathcal{D}_{pub}$ to $\mathcal{D}_{priv}$, compared to fine-tuning all layers (0-15).

We then computed the singular values of $\Delta W_q = W_{q,\text{finetuned}} - W_{q,\text{pretrained}}$ ($\Delta W_k, \Delta W_v$ are defined with respect to $k, v$ matrices). As shown in (b) of Fig. 5, $\Delta W$ for K,V matrices are correlated possibly because

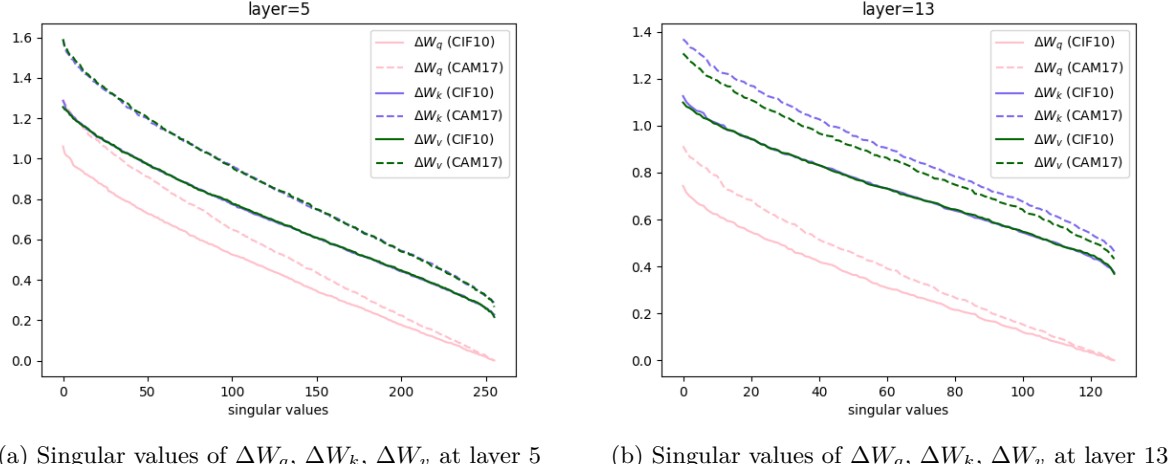

(a) Singular values of $\Delta W_q$, $\Delta W_k$, $\Delta W_v$ at layer 5     (b) Singular values of $\Delta W_q$, $\Delta W_k$, $\Delta W_v$ at layer 13

Figure 6: Singular values of $\Delta W_q$, $\Delta W_k$, $\Delta W_v$ in cross-attention modules in the conditional LDMs fine-tuned for CIFAR10 (CIF10) and that for Camelyon17 (CAM17), at $\epsilon = 10$ with $\delta = 10^{-5}$ for CIFAR10 and $\delta = 3 \cdot 10^{-6}$ for Camelyon17. Dotted lines represent CAM17, while solid lines represent CIF10. Purple and green solid lines are overlapping in both plots.

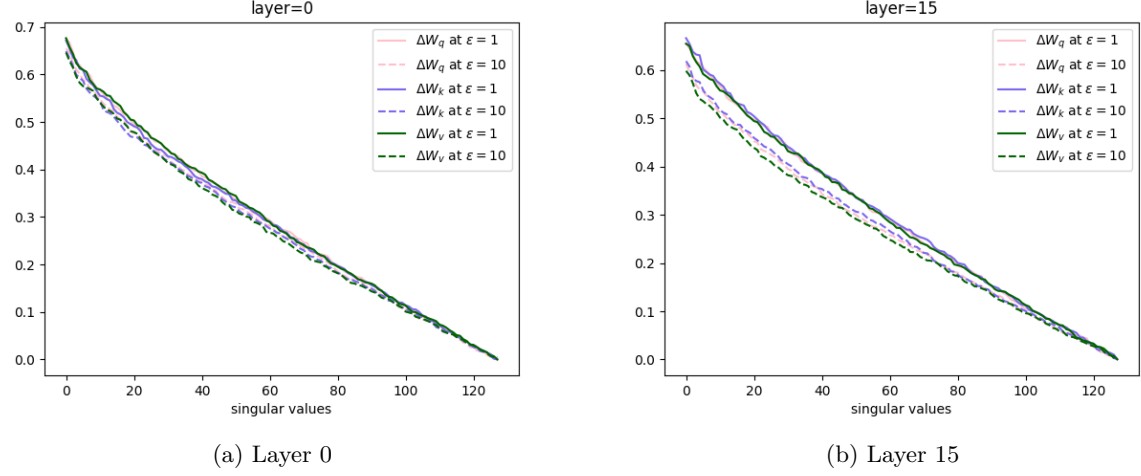

(a) Layer 0                                    (b) Layer 15

Figure 7: Singular values of $\Delta W_q$, $\Delta W_k$, $\Delta W_v$ in self-attention modules at varying layers of an unconditional LDM, fine-tuned for CelebA64, at $\epsilon = 10$ and $\epsilon = 1$. Dotted lines represent the model fine-tuned at $\epsilon = 10$, and solid lines that at $\epsilon = 1$.

the role of $W_k$ and $W_v$ is to pick which class embeddings are more useful (both are multiplied by the class embedding as in eq. 4), their fine-tuned values remain similar. The singular values fall off fast in the case of $\Delta W_q$, while those of $\Delta W_k$ and $\Delta W_v$ fall off slowly and even the smallest singular values are far from zero. Similarly, in Fig. 6, the singular values of fine-tuned model for Camelyon17 seem to decay faster than those for CIFAR10. However, the smallest singular values at the later layer are again far from zero. This might imply low rank approximation to these matrices can cause losing information. This observation is consistent with the performance of DP-LoRA lagging that of DP-LDM shown in Table 5 and Table 6.

Fig. 7 shows the singular values of $\Delta W_q$, $\Delta W_k$, $\Delta W_v$ in the unconditional LDM fine-tuned for CelebA64. There is only a slight difference between the singular values of the model fine-tuned at $\epsilon = 1$ and the model fine-tuned at $\epsilon = 10$. This could be explained by using a fixed random seed during training. As the random seed determines the order of training batches as well as the direction of the random noise injected by DP-SGD, each model converges to similar areas of the parameter space. Thus, differences in the fine-tuned weight matrices arise from the different magnitudes of noise added. On the other hand, this is no longer the case,

when the target datasets change like in Fig. 6, or when the ablation setting (which layers to be fine-tuned) changes like in Fig. 5, even if we set the seed number to be the same across different settings.

## 6 Conclusion

In *Differentially Private Latent Diffusion Models* (DP-LDM), we utilize DP-SGD to fine-tune only the attention modules (and embedders for conditioned generation) of the pretrained LDM at varying layers with privacy-sensitive data. We demonstrate that our method is capable of generating quality images in various scenarios. We perform an in-depth analysis of the ablation of DP-LDM to explore the strategy for reducing parameters for more applicable training of DP-SGD. Based on our promising results, we conclude that fine-tuning LDMs is an efficient and effective framework for DP generative learning. We hope our results can contribute to future research in DP data generation, considering the rapid advances in diffusion-based generative modelling.

## Broader Impact Statement

As investigated in Carlini et al. (2023), diffusion models can memorize individual images from the training data set and generate an identical synthetic image to the training image. Aiming to impact society positively, we provide a method to fine-tune latent diffusion models with differential privacy guarantees so that the fine-tuned models do not generate identical images to those in the private data.

Our method relies on public data for better scalability of differential privacy, which needs some attention. As Tramèr et al. (2022) pointed out, public data themselves may still contain sensitive information. From our perspective and as many other DP generative modelling papers also noticed, auxiliary public data still emerges as the most promising option for attaining satisfactory utility, for the models at this large scale. We hope for better-curated public datasets to be available in the near future.

During the review process, one of the reviewers mentioned that generative tools are becoming more and more realistic and of general availability, with the risk of generating harmful content, especially when generating human-related content. Broadly speaking, our method does not stop malicious users from creating harmful content, as the diffusion model will generate images based on any input prompt given by the malicious user. However, what our method can do is protect the privacy of the individuals whose data was in the private dataset, meaning the faces in the private dataset will not appear when generating images from the DP fine-tuned models.

To encourage reproducible research practices, our code is available at https://github.com/ParkLabML/DP-LDM with detailed instructions. And all the hyperparameters are discussed in detail in Appendix Sec. B.

## Limitations

While the paper proposes a technique to counter-attack the privacy risks of diffusion models, the evaluation has not been done against existing privacy attacks. So we know that the diffusion model is private "under the definition of DP", but we caution practitioners against making an assumption that a DP image generative model protects private data against all possible attack methods.

This may not be a direct limitation of our work itself, but it is restrictive that there is no agreement within the community on which definition of privacy should be used to assess the privacy of generated images. Carlini et al. (2023) provided a single example of exploitation, but it remains unclear what aspects we aim to protect in generated data. Unfortunately, the privacy parameters $\epsilon$ and $\delta$ in our method are especially difficult to interpret in the image domain. While private images may be protected, it is unclear whether perceptually similar images enjoy the same level of privacy. Further work is required to suggest more appropriate definitions of privacy for practitioners to employ to avoid providing a vague sense of privacy in the image generation domain.

**Acknowledgments**

We thank our anonymous reviewers for their constructive feedback, which has helped significantly improve our manuscript. We thank the Digital Research Alliance of Canada (Compute Canada) for its computational resources and services. M. Liu was funded by the Canada Graduate Scholarships — Master's program of the Natural Sciences and Engineering Research Council of Canada (NSERC). S. Lyu, M. Vinaroz, and M. Park were supported in part by the Natural Sciences and Engineering Research Council of Canada (NSERC) and the Canada CIFAR AI Chairs program. M. Park was also partially funded by Novo Nordisk Fonden RECUIT grant no.0065800 during her stay at the Technical University of Denmark.

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

# Appendix

## A  Additional Experiments

### A.1  Scaling factor effect in pre-training the autoencoder

In Table 8, we provide FID results after pre-training the autoencoder with Imagenet dataset for different scaling factors $f$ and number of channels.

|         | # channels | | |
|---------|------|------|------|
|         | 128  | 64   | 32   |
| $f = 2$ | **27.6** | 36.4 | 46.8 |
| $f = 4$ | 32.9 | 51.0 | 83.5 |

Table 8: FID scores (lower is better) for pre-trained autoencoders with different $f$ and number of channels.

### A.2  Transferring from EMNIST to MNIST distribution

Here are the details when we compare DP-LDM to existing methods with the most common DP settings $\epsilon = 1, 10$ and $\delta = 10^{-5}$ in Table 9.

|                |          | DP-LDM (Ours) | DP-DM | DP-Diffusion | DP-HP | PEARL | DPGANr |
|----------------|----------|---------------|-------|--------------|-------|-------|--------|
| $\epsilon = 10$ | CNN      | 97.4± 0.1     | **98.1** | -         | -     | 78.8  | 95.0   |
|                | WRN      | 97.5± 0.0     | -     | **98.6**     | -     | -     | -      |
| $\epsilon = 1$ | CNN      | **95.9± 0.1** | 95.2  | -            | 81.5  | 78.2  | 80.1   |
|                | # params | **0.8M**      | 1.75M | 4.2M         | -     | -     | -      |
|                | GPU Hours | **10h**      | 192h  | -            | -     | -     | -      |

Table 9: Downstream accuracies by CNN, MLP and WRN-40-4, evaluated on the generated MNIST data samples. We compare our results with existing work DPDM (Dockhorn et al., 2023), DP-Diffusion (Ghalebikesabi et al., 2023), PEARL (Liew et al., 2022b), DPGANr (Bie et al., 2022), and DP-HP (Vinaroz et al., 2022). The GPU hours is for DP training only. The GPU hours for pretraining steps of our method are present in Table 17 and Table 18.

#### A.2.1  Choosing public dataset with DP constraint

FID scores are commonly used for measuring the similarity of two dataset. It first uses a pre-trained neural network (such as InceptionV3) to extract features from both datasets; then fits two Gaussian distributions to both datasets respectively, via computing the mean and covariance of the feature representations for both of them; then use the means and covariances to calculate the Fréchet distance. Following Park et al. (2017), we computed the FID scores between public data and private data in a differentially private manner. I.e. consider a data matrix by $X$, where $n$ i.i.d. observations in a privacy-sensitive dataset are stacked in each row of $X$. We denote each observation of this dataset by $\mathbf{x}_i \in \mathbb{R}^d$. Hence, $X \in \mathbb{R}^{n \times d}$. We denote the inception feature given a datapoint $\mathbf{x}_i$ by $\boldsymbol{\phi}(\mathbf{x}_i)$. We further denote the mean and the covariance of the inception features, computed on a public dataset, by $\boldsymbol{\mu}_0$ and $\Sigma_0$. Similarly, we denote those computed on a privacy-sensitive dataset by $\boldsymbol{\mu}, \Sigma$. The non-DP computation of FID is given by the following formula (notations are only used in this subsection):

$$\mathbf{FID} = \|\boldsymbol{\mu}_0 - \boldsymbol{\mu}\|_2^2 + \operatorname{tr}\left[\Sigma_0 + \Sigma - 2\left(\Sigma_0 \Sigma\right)^{\frac{1}{2}}\right]$$

We will need to privatize $\mu$ and $\Sigma$. Following Park et al. (2017), we privatize the mean vector using a $(\epsilon_1, \delta_1)$-DP Gaussian mechanism. Let us first recall the definition of $\boldsymbol{\mu}$ given by :

$$\boldsymbol{\mu} := \frac{1}{n} \sum_{i=1}^n \phi(\mathbf{x}_i).$$

Assuming $\|\phi(\mathbf{x}_i)\| \leq 1$ for any $i$, the sensitivity of the mean vector denoted by $\Delta_{\boldsymbol{\mu}}$ is :

$$
\begin{aligned}
\Delta_{\boldsymbol{\mu}} &= \max_{\mathbf{x}_j, \mathbf{x}_j'} \| \frac{1}{n} \big( \sum_{i=1, i \neq j}^n \phi(\mathbf{x}_i) + \phi(\mathbf{x}_j) \big) - \frac{1}{n} \big( \sum_{i=1, i \neq j}^n \phi(\mathbf{x}_i) + \phi(\mathbf{x}_j') \big) \| \\
&= \frac{1}{n} \max_{\mathbf{x}_j, \mathbf{x}_j'} \| \phi(\mathbf{x}_j) - \phi(\mathbf{x}_j') \| \\
&\leq \frac{1}{n} \cdot 2 \cdot \| \phi(\mathbf{x}_i) \| \\
&= 2/n
\end{aligned}
$$

i.e., bounded by $2/n$, when using the replacement definition of differential privacy (it is $1/n$ when using the inclusion/exclusion definition of DP). Hence,

$$\tilde{\boldsymbol{\mu}} := \boldsymbol{\mu} + \mathbf{n}_1, \tag{5}$$

where $\mathbf{n}_1$ is drawn from $\mathcal{N}(0, \Delta_{\boldsymbol{\mu}}^2 \sigma_1^2 I)$. Here, $\sigma_1$ is a function of the privacy level given by $(\epsilon_1, \delta_1)$-DP. The exact relationship between $\sigma_1$ and $(\epsilon_1, \delta_1)$ varies depending on how to compute the DP bound. We use the bound introduced in the analytic Gaussian mechanism in https://github.com/yuxiangw/autodp.

Recall the definition of covariance given by:

$$\Sigma = \frac{1}{n} X^T X - \boldsymbol{\mu} \boldsymbol{\mu}^T.$$

Since we have a privatized mean from above, we need to privatize the second-moment matrix $\frac{1}{n} X^T X = M_{\text{sec}}$ to privatize the covariance matrix.

As before, assuming $\|\phi(\mathbf{x}_i)\| \leq 1, \forall i$, the sensitivity of the second moment matrix denoted by $\Delta_{M_{\text{sec}}}$ is :

$$
\begin{aligned}
\Delta_{M_{\text{sec}}} &= \max_{\phi(\mathbf{x}_j), \phi(\mathbf{x}_j')} \| \frac{1}{n} \big( \sum_{i=1, i \neq j}^n \phi(\mathbf{x}_i) \phi(\mathbf{x}_i^T) + \phi(\mathbf{x}_j) \phi(\mathbf{x}_j^T) \big) - \frac{1}{n} \big( \sum_{i=1, i \neq j}^n \phi(\mathbf{x}_i) \phi(\mathbf{x}_i^T) + \phi(\mathbf{x}_j') \phi(\mathbf{x}_j'^T) \big) \|_F \\
&= \frac{1}{n} \max_{\phi(\mathbf{x}_j), \phi(\mathbf{x}_j')} \| \phi(\mathbf{x}_j) \phi(\mathbf{x}_j^T) - \phi(\mathbf{x}_j') \phi(\mathbf{x}_j'^T) \|_F \\
&\leq \frac{2}{n} \max_{\phi(\mathbf{x}_j)} \| \phi(\mathbf{x}_j) \phi(\mathbf{x}_j^T) \|_F && \text{(WLOG, say } \| \phi(\mathbf{x}_j') \phi(\mathbf{x}_j'^T) \|_F \leq \| \phi(\mathbf{x}_j) \phi(\mathbf{x}_j^T) \|_F) \\
&= \frac{2}{n} \max_{\phi(\mathbf{x}_j)} \sqrt{\sum_{i=1}^d \sum_{k=1}^d |\alpha_i \alpha_k|^2} && \text{(Say } \phi(\mathbf{x}_j) = [\alpha_1, \cdots, , \alpha_d] \text{ as a column vector)} \\
&= \frac{2}{n} \max_{\phi(\mathbf{x}_j)} \sqrt{\sum_{i=1}^d \big( \alpha_i^2 \cdot \sum_{k=1}^d \alpha_k^2 \big)} \\
&= \frac{2}{n} \max_{\phi(\mathbf{x}_j)} \sqrt{\big( \sum_{i=1}^d \alpha_i^2 \big) \cdot \big( \sum_{k=1}^d \alpha_k^2 \big)} \\
&= \frac{2}{n} \max_{\phi(\mathbf{x}_j)} \sqrt{\| \phi(\mathbf{x}_j) \|_2^2 \cdot \| \phi(\mathbf{x}_j) \|_2^2} \\
&\leq \frac{2}{n}
\end{aligned}
$$

| $\epsilon$ | 0.1 | 0.5 | 1 | 2 | 5 | 10 | $\epsilon$ | 0.1 | 0.5 | 1 | 2 |
|---|---|---|---|---|---|---|---|---|---|---|---|
| SVHN | **78.9930** | **17.3465** | 8.9223 | 4.7426 | 2.1582 | 1.3368 | SVHN | 0.2675 | 0.2662 | 0.2663 | 0.2663 |
| KMNIST | 79.9224 | 17.5794 | 9.0605 | 4.6092 | 1.9300 | 1.0794 | KMNIST | 0.0489 | 0.0459 | 0.0457 | 0.0457 |
| **EMNIST** | 80.4267 | 17.3617 | **8.9058** | **4.4561** | **1.8926** | **1.0242** | **EMNIST** | **0.0228** | **0.0201** | **0.0201** | **0.0197** |

Table 10: Left : Perturbed FIDs when privatizing both mean and covariance. Note $\epsilon$ listed is the sum of $\epsilon_\mu + \epsilon_\Sigma$, we take $\epsilon_\mu = \epsilon_\Sigma$. Right : Perturbed mean differences when privatizing mean only. $\epsilon$ listed is $\epsilon_\mu$

i.e., bounded by $2/n$, when using the replacement definition of differential privacy (it is $1/n$ when using the inclusion/exclusion definition of DP). See eq(17) in Park et al. (2017) for derivation of the sensitivity.

Given a privacy budget $(\epsilon_2, \delta_2)$-DP assigned to this privatization step, which gives us a corresponding privacy parameter $\sigma_2$, we first draw noise $\mathbf{n}_2$ from $\mathcal{N}(0, \Delta^2_{M_{\text{sec}}} \sigma_2^2 I_{d(d+1)/2})$ . We then add this noise to the upper triangular part of the matrix, including the diagonal component. To ensure the symmetry of the perturbed second-moment matrix $\widetilde{M_{\text{sec}}}$, we copy the noise added to the upper triangular part and add one by one to the lower triangular part. We then obtain

$$\tilde{\Sigma} = \widetilde{M_{\text{sec}}} - \tilde{\boldsymbol{\mu}}\tilde{\boldsymbol{\mu}}^T. \tag{6}$$

Note $\tilde{\Sigma}$ may not be positive definite. In such a case, we can project the negative eigenvalue to some small value close to zero to guarantee the positive definite property of the covariance matrix. This is safe to do, as DP is post-processing invariant.

Using the aforementioned privatized mean given in eq. 5 and covariance given in eq. 6, we can compute the final FID score, given by

$$\mathbf{DP\text{-}FID} = \|\boldsymbol{\mu}_0 - \tilde{\boldsymbol{\mu}}\|_2^2 + \text{tr}\left[\Sigma_0 + \tilde{\Sigma} - 2\left(\Sigma_0\tilde{\Sigma}\right)^{\frac{1}{2}}\right],$$

which is $(\epsilon_1 + \epsilon_2, \delta_1 + \delta_2)$-DP.

Following the above method, we compute the DP-FID scores of SVHN, KMNIST, and EMNIST, with respect to private dataset MNIST. We sampled 60k data from each public dataset candidates to do a fair comparison, the results are in Table 10 left. One could consider only privatizing the mean only, and the results are in Table 10 right. Based on these results, we choose EMNIST as a public dataset.

### A.2.2   Additional experiments with SVHN and KMNIST

To verify our choice of EMNIST emperically, we also did ablation experiments on SVHN and KMNIST under the same privacy condition $\epsilon = 10, \delta = 10^{-5}$ to compare with EMNIST, the results are illustrated in Table 11.

| Dataset pair | CNN accuracy |
|---|---|
| (SVHN, MNIST) | 94.3 |
| (KMNIST, MNIST) | 96.3 |
| **(EMNIST, MNIST)** | **97.4** |

Table 11: We also pretrained LDMs using SVHN and KMNIST then fine-tuned with MNIST, and list the CNN accuracy here respectively.

### A.2.3   Ablation experiments for MNIST

There are 7 attention modules in the UNet structure for MNIST, 1-2 are in input_blocks, 3 is in middle_block, 4-7 are in out_blocks as illustrated in Fig. 8. Modules in blue are frozen during fine-tuning. Parameters of condition embedder is always trained.

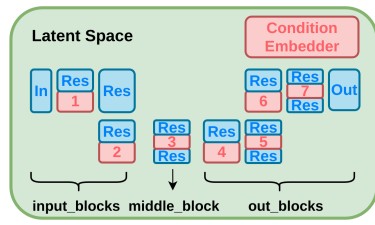

Figure 8: UNet Structure for MNIST.

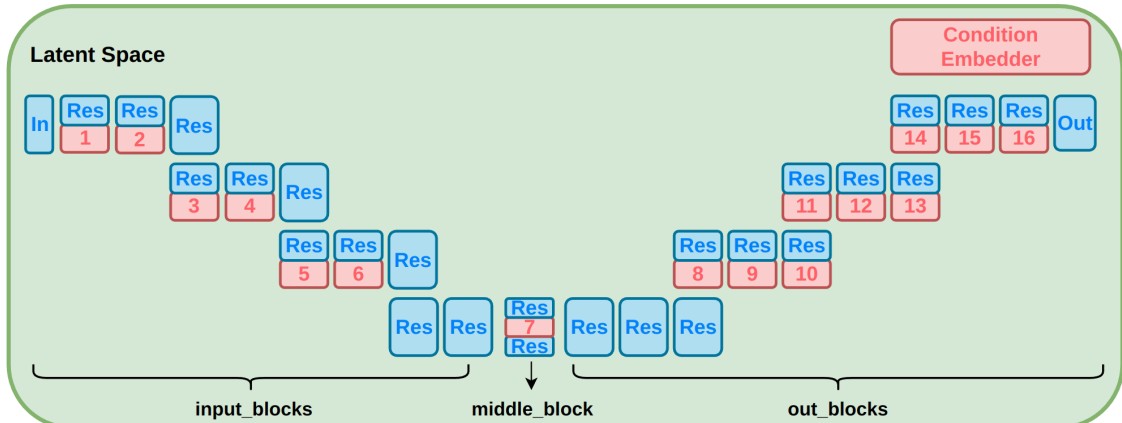

Figure 9: UNet Structure for CIFAR-10

We consider fine tune only $i$-th to 7th attention modules to reduce more trainable parameters. Results for $\epsilon = 10, \delta = 10^{-5}$ are listed in Table 12. The best results is achieved when fine tune with 4-7 attention blocks, which means out_blocks are more important than others during training.

|  | 1-7(all) | 2-7 | 3-7 | 4-7 | 5-7 |
|---|---|---|---|---|---|
| CNN | 97.3 | 97.3 | 90 | **97.4** | 97.3 |
| # of trainable params | 1.6M | 1.5M | 1.2M | 0.8M | 0.5M |
| out of 4.6M total params | (34.3%) | (32.4%) | (25.2%) | (18.0%) | (10.9%) |

Table 12: CNN accuracy and number of trainable parameters for MNIST ablation experiments with varying number of fine-tuning layers. Privacy condition is set to $\epsilon = 10, \delta = 10^{-5}$.

### A.3 Transferring from Imagenet to CIFAR10 distribution

Here, we provide the results for ablation experiments to test the performance of DP-LDM when fine-tuning only certain attention modules inside the pre-trained model and keeping the rest of the parameters fixed. There are 16 attention modules in total as illustrated in Fig. 9. Table 13 shows the FID obtained for $\epsilon = 1, 5, 10$ and $\delta = 10^{-5}$ for the different number of attention modules fine-tuned. The results show that fixing up to the first half of the attention layers in the LDM has a positive effect in terms of the FID (the lower the better) in our model.

| DP-LDM | $\epsilon = 10$ | $\epsilon = 5$ | $\epsilon = 1$ |
|---|---|---|---|
| 1-16 layers | $25.8 \pm 0.3$ | $29.9 \pm 0.2$ | $33.0 \pm 0.3$ |
| 5 - 16 layers | $15.7 \pm 0.3$ | $21.2 \pm 0.2$ | $28.9 \pm 0.2$ |
| 9 - 16 layers | $\mathbf{8.4 \pm 0.2}$ | $\mathbf{13.4 \pm 0.4}$ | $\mathbf{22.9 \pm 0.5}$ |
| 13 - 16 layers | $12.3 \pm 0.2$ | $18.5 \pm 0.2$ | $25.2 \pm 0.5$ |

Table 13: FID scores (lower is better) for synthetic CIFAR-10 data with varying the number of fine-tuning layers and privacy guarantees. **Top row (1-16 layers):** Fine-tuning all attention modules. **Second row (5-16 layers):** Keep first 4 attention modules fixed and fine-tuning from 5 to 16 attention modules. **Third row (9-16 layers):** Keep first 8 attention modules fixed and fine-tuning from 9 to 16 attention modules. **Bottom row (13-16 layers):** Keep first 12 attention modules fixed and fine-tuning from 13 to 16 attention modules.

We also report the different hyper-parameter settings used in ablation experiments in table Table 14.

| | | $\epsilon = 10$ | $\epsilon = 5$ | $\epsilon = 1$ |
|---|---|---|---|---|
| 1-16 layers (24.4M parameters) | batch size | 1000 | 2000 | 1000 |
| | clipping norm | $10^{-5}$ | $10^{-5}$ | $10^{-3}$ |
| | learning rate | $10^{-6}$ | $10^{-6}$ | $10^{-6}$ |
| | epochs | 30 | 30 | 10 |
| 5-16 layers (20.8M parameters) | batch size | 5000 | 5000 | 2000 |
| | clipping norm | $10^{-6}$ | $10^{-5}$ | $10^{-3}$ |
| | learning rate | $10^{-6}$ | $10^{-6}$ | $10^{-5}$ |
| | epochs | 50 | 50 | 10 |
| 9-16 layers (10.2M parameters) | batch size | 2000 | 2000 | 5000 |
| | clipping norm | $10^{-6}$ | $10^{-6}$ | $10^{-2}$ |
| | learning rate | $10^{-6}$ | $10^{-6}$ | $10^{-6}$ |
| | epochs | 30 | 30 | 10 |
| 13-16 layers (4M parameters) | batch size | 2000 | 2000 | 2000 |
| | clipping norm | $10^{-6}$ | $10^{-6}$ | $10^{-2}$ |
| | learning rate | $10^{-6}$ | $10^{-6}$ | $10^{-6}$ |
| | epochs | 30 | 30 | 10 |

Table 14: DP-LDM hyper-parameter setting on CIFAR-10 for different ablation experiments.

| | ResNet9 | WRN-40-4 |
|---|---|---|
| learning rate | 0.5 | 0.1 |
| batch size | 512 | 1000 |
| epochs | 10 | 10000 |
| optimizer | SGD | SGD |
| label smoothing | 0.1 | 0.0 |
| weight decay | $5 \cdot 10^{-4}$ | $5 \cdot 10^{-4}$ |
| momentum | 0.9 | 0.9 |

Table 15: Hyperparameters for downstream classification ResNet9 and WRN-40-4 trained on CIFAR10 synthetic data

Table 15 shows the hyper-parameters used during training ResNet9 and WRN-40-4 downstream classifiers on CIFAR10 synthetic samples.

## A.4 Transferring from Imagenet to CelebA32

We also apply our model in the task of generating $32 \times 32$ CelebA images. The same pretrained autoencoder as our CIFAR-10 experiments in Section 5.1 is used, but since this experiment is for unconditional generation, we are unable to re-use the LDM. A new LDM is pretrained on Imagenet without class conditioning information, and then fine-tuned on CelebA images scaled and cropped to $32 \times 32$ resolution. Our FID results for $\delta = 10^{-6}$, $\epsilon = 1, 5, 10$ are summarized in Table 16. We achieve similar results to DP-MEPF for $\epsilon = 5$ and $\epsilon = 10$. As with our results at $64 \times 64$ resolution, our LDM model does not perform as well in higher privacy settings ($\epsilon = 1$). Sample images are provided in Fig. 10

## A.5 Transferring from Imagenet to Camelyon17-WILDS

Camelyon17-WILDS is part of the WILDS benchmark suite of datasets, containing $455,954$ medical images at $96 \times 96$ resolution. The downstream task is to determine whether the center $32 \times 32$ patch of the image contains any tumor pixels. In our experiment, we crop the image so that only the center $32 \times 32$ patch is passed to the model. We begin with the same pretrained autoencoder and LDM as in our CIFAR-10 experiments (Section 5.1), and fine-tune on Camelyon17-WILDS. We then generate $25,000$ images conditioned

|  | $\epsilon = 10$ | $\epsilon = 5$ | $\epsilon = 1$ |
|---|---|---|---|
| **DP-LDM** (ours, average) | $16.2 \pm 0.2$ | $16.8 \pm 0.3$ | $25.8 \pm 0.9$ |
| **DP-LDM** (ours, best) | **16.1** | **16.6** | 24.6 |
| DP-MEPF ($\phi_1$) | 16.3 | 17.2 | **17.2** |
| DP-GAN (pre-trained) | 58.1 | 66.9 | 81.3 |
| DPDM (no public data) | 21.2 | - | 71.8 |
| DP Sinkhorn (no public data) | 189.5 | - | - |

Table 16: CelebA FID scores (lower is better) for images of resolution $32 \times 32$ comparing with results from DPDM (Dockhorn et al., 2023), DP Sinkhorn (Cao et al., 2021), and DP-MEPF (Harder et al., 2023).

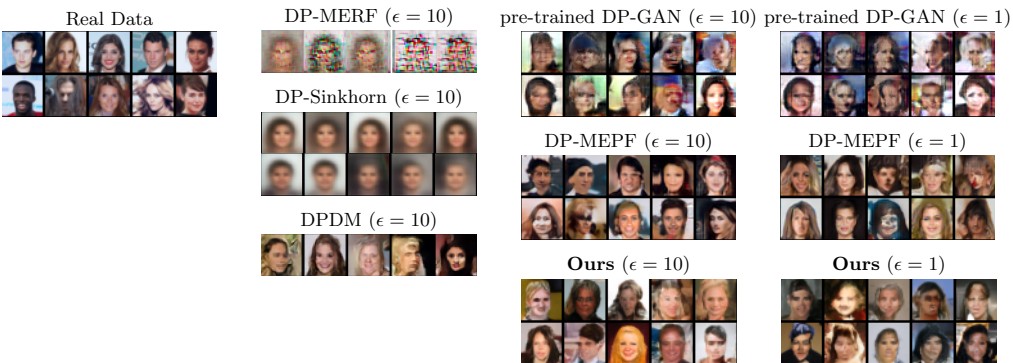

Figure 10: Synthetic $32 \times 32$ CelebA samples generated at different levels of privacy. Samples for DP-MERF and DP-Sinkhorn are taken from (Cao et al., 2021), DPDM samples are taken from (Dockhorn et al., 2023), and DP-MEPF samples are taken from (Harder et al., 2023).

on each of the two classes, and combine them to create a synthetic dataset of $50,000$ images. Finally, the synthetic dataset is used to train a WRN-40-4 classifier.

## B  Hyperparameters

Here we provide an overview of the hyperparameters of the pretrained autoencoder in Table 17, hyperparameters of the pretrained diffusion models in Table 18. Note that *base learning rate* is the one set in the yaml files. The real learning rate passed to the optimizer is *accumulate_grad_batches × num_gpus × batch size × base learning rate*.

|  | EMNIST (to MNIST) | ImageNet (to CIFAR10) | ImageNet (to CelebA 32) | ImageNet (to CelebA 64) |
|---|---|---|---|---|
| Input size | 32 | 32 | 32 | 64 |
| Latent size | 4 | 16 | 16 | 32 |
| $f$ | 8 | 2 | 2 | 2 |
| $z$-shape | $4 \times 4 \times 3$ | $16 \times 16 \times 3$ | $16 \times 16 \times 3$ | $64 \times 64 \times 3$ |
| Channels | 128 | 128 | 128 | 192 |
| Channel multiplier | [1,2,3,5] | [1,2] | [1,2] | [1,2] |
| Attention resolutions | [32,16,8] | [16, 8] | [16, 8] | [16,8] |
| num_res_blocks | 2 | 2 | 2 | 2 |
| Batch size | 50 | 16 | 16 | 16 |
| Base learning rate | $4.5 \times 10^{-6}$ | $4.5 \times 10^{-6}$ | $4.5 \times 10^{-6}$ | $1.0 \times 10^{-6}$ |
| Learning rate | $4.5 \times 10^{-4}$ | $1.4 \times 10^{-4}$ | $1.4 \times 10^{-4}$ | $1.4 \times 10^{-4}$ |
| Epochs | 50 | 2 | 2 | - |
| GPU(s) | 1 NVIDIA V100 | 1 NVIDIA RTX A4000 | 1 NVIDIA RTX A4000 | 1 NVIDIA V100 |
| Time | 8 hours | 1 day | 1 day | 1 day |

Table 17: Hyperparameters for the pretrained autoencoders for different datasets.

| | EMNIST (to MNIST) | ImageNet (to CIFAR10) | ImageNet (to CelebA 32) | ImageNet (to CelebA64) |
|---|---|---|---|---|
| input size | 32 | 32 | 32 | 64 |
| latent size | 4 | 16 | 16 | 32 |
| $f$ | 8 | 2 | 2 | 2 |
| $z$-shape | $4 \times 4 \times 3$ | $16 \times 16 \times 3$ | $16 \times 16 \times 3$ | $32 \times 32 \times 3$ |
| channels | 64 | 128 | 192 | 192 |
| channel multiplier | [1,2] | [1,2,2,4] | [1,2,4] | [1,2,4] |
| attention resolutions | [1,2] | [1,2,4] | [1,2,4] | [1,2,4] |
| num_res_blocks | 1 | 2 | 2 | 2 |
| num_heads | 2 | 8 | - | 8 |
| num_head_channels | - | - | 32 | - |
| batch size | 512 | 500 | 384 | 256 |
| base learning rate | $5 \times 10^{-6}$ | $1 \times 10^{-6}$ | $5 \times 10^{-7}$ | $1 \times 10^{-6}$ |
| learning rate | $2.6 \times 10^{-3}$ | $5 \times 10^{-4}$ | $2 \times 10^{-4}$ | $2.6 \times 10^{-4}$ |
| epochs | 120 | 30 | 13 | 40 |
| # trainable parameters | 4.6M | 90.8M | 162.3M | 72.2M |
| GPU(s) | 1 NVIDIA V100 | 1 NVIDIA RTX A4000 | 1 NVIDIA V100 | 1 NVIDIA V100 |
| time | 6 hours | 7 days | 30 hours | 10 days |
| use_spatial_transformer | True | True | False | False |
| cond_stage_key | class_label | class_label | - | - |
| conditioning_key | crossattn | crossattn | - | - |
| num_classes | 26 | 1000 | - | - |
| embedding dimension | 5 | 512 | - | - |
| transformer depth | 1 | 1 | - | - |

Table 18: Hyperparameters for the pretrained diffusion models for different datasets.

Table 19 shows the hyperparameters we used for fine-tuning on MNIST. Table 20 shows the hyperparameters we used for CelebA32. Table 21 shows the hyperparameters we used for CelebA64. Table 23 shows the hyperparameters we used for text-to-image CelebAHQ generation. Table 24 shows the hyperparmeters we used for class-conditioned CelebAHQ generation.

| | $\epsilon = 10$ | $\epsilon = 1$ |
|---|---|---|
| batch size | 2000 | 2000 |
| base learning rate | $5 \times 10^{-7}$ | $6 \times 10^{-7}$ |
| learning rate | $1 \times 10^{-3}$ | $1.2 \times 10^{-3}$ |
| epochs | 200 | 200 |
| clipping norm | 0.01 | 0.001 |
| noise scale | 1.47 | 9.78 |
| ablation | 4 | -1 |
| num of params | 0.8M | 1.6M |
| use_spatial_transformer | True | True |
| cond_stage_key | class_label | class_label |
| conditioning_key | crossattn | crossattn |
| num_classes | 26 | 26 |
| embedding dimension | 13 | 13 |
| transformer depth | 1 | 1 |
| train_condition_only | True | True |
| attention_flag | spatial | spatial |
| # condition params | 338 | 338 |

Table 19: Hyperparameters for fine-tuning diffusion models with DP constraints $\epsilon = 10, 1$ and $\delta = 10^{-5}$ on MNIST. The "ablation" hyperparameter determines which attention modules are fine-tuned, where a value of $i$ means that the first $i - 1$ attention modules are frozen and others are trained. Setting "ablation" to $-1$ (default) fine-tunes all attention modules.

|  | $\epsilon = 10$ | $\epsilon = 5$ | $\epsilon = 1$ |
|---|---|---|---|
| batch size | 8192 | 8192 | 2048 |
| base learning rate | $5 \times 10^{-7}$ | $5 \times 10^{-7}$ | $5 \times 10^{-7}$ |
| learning rate | $4 \times 10^{-3}$ | $4 \times 10^{-3}$ | $1 \times 10^{-3}$ |
| epochs | 20 | 20 | 20 |
| clipping norm | $5.0 \times 10^{-4}$ | $5.0 \times 10^{-4}$ | $5.0 \times 10^{-4}$ |
| ablation | -1 | -1 | -1 |
| use_spatial_transformer | False | False | False |
| cond_stage_key | - | - | - |
| conditioning_key | - | - | - |
| num_classes | - | - | - |
| embedding dimension | - | - | - |
| transformer depth | - | - | - |
| train_attention_only | True | True | True |

Table 20: Hyperparameters for fine-tuning diffusion models with DP constraints $\epsilon = 10, 5, 1$ and $\delta = 10^{-6}$ on CelebA32.

|  | $\epsilon = 10$ | $\epsilon = 5$ | $\epsilon = 1$ |
|---|---|---|---|
| batch size | 8192 | 8192 | 8192 |
| base learning rate | $1 \times 10^{-7}$ | $1 \times 10^{-7}$ | $1 \times 10^{-7}$ |
| learning rate | $8.2 \times 10^{-4}$ | $8.2 \times 10^{-4}$ | $8.2 \times 10^{-4}$ |
| epochs | 70 | 70 | 70 |
| clipping norm | $5.0 \times 10^{-4}$ | $5.0 \times 10^{-4}$ | $5.0 \times 10^{-4}$ |
| ablation | -1 | -1 | -1 |
| use_spatial_transformer | False | False | False |
| cond_stage_key | - | - | - |
| conditioning_key | - | - | - |
| num_classes | - | - | - |
| embedding dimension | - | - | - |
| transformer depth | - | - | - |
| train_attention_only | True | True | True |

Table 21: Hyperparameters for fine-tuning diffusion models with DP constraints $\epsilon = 10, 5, 1$ and $\delta = 10^{-6}$ on CelebA64.

|  | $\epsilon = 10$ |
|---|---|
| batch size | 20000 |
| base learning rate | $1.0 \times 10^{-7}$ |
| learning rate | $2.0 \times 10^{-3}$ |
| epochs | 30 |
| clipping norm | $1.0 \times 10^{-6}$ |
| ablation | 13 |
| use_spatial_transformer | True |
| cond_stage_key | class_label |
| conditioning_key | crossattn |
| num_classes | 1001 |
| embedding dimension | 512 |
| transformer depth | 1 |
| train_condition_only | True |
| attention_flag | spatial |
| # condition params | $512, 512$ |

Table 22: Hyperparameters for fine-tuning diffusion models with DP constraint $\epsilon = 10$ and $\delta = 3 \times 10^{-6}$ on Camelyon17-WILDS.

|  | $\epsilon = 10$ | $\epsilon = 1$ |
|---|---|---|
| batch size | 256 | 256 |
| base learning rate | $1 \times 10^{-7}$ | $1 \times 10^{-7}$ |
| learning rate | $2.6 \times 10^{-5}$ | $2.6 \times 10^{-5}$ |
| epochs | 10 | 10 |
| clipping norm | 0.01 | 0.01 |
| noise scale | 0.55 | 1.46 |
| ablation | -1 | -1 |
| num of params | 280M | 280M |
| use_spatial_transformer | True | True |
| cond_stage_key | caption | caption |
| context_dim | 1280 | 1280 |
| conditioning_key | crossattn | crossattn |
| transformer depth | 1 | 1 |

Table 23: Hyperparameters for fine-tuning diffusion models with DP constraints $\epsilon = 10, 1$ and $\delta = 10^{-5}$ on text-conditioned CelebAHQ.

|  | $\epsilon = 10$ | $\epsilon = 5$ | $\epsilon = 1$ |
|---|---|---|---|
| batch size | 2048 | 2048 | 2048 |
| base learning rate | $1 \times 10^{-7}$ | $1 \times 10^{-7}$ | $1 \times 10^{-7}$ |
| learning rate | $2.0 \times 10^{-4}$ | $2.0 \times 10^{-4}$ | $2.0 \times 10^{-4}$ |
| epochs | 50 | 50 | 50 |
| clipping norm | $5.0 \times 10^{-4}$ | $5.0 \times 10^{-4}$ | $5.0 \times 10^{-4}$ |
| ablation | -1 | -1 | -1 |
| use_spatial_transformer | True | True | True |
| cond_stage_key | class_label | class_label | class_label |
| context_dim | 512 | 512 | 512 |
| conditioning_key | crossattn | crossattn | crossattn |
| transformer depth | 1 | 1 | 1 |

Table 24: Hyperparameters for fine-tuning diffusion models with DP constraints $\epsilon = 10, 5, 1$ and $\delta = 10^{-5}$ on class-conditional CelebAHQ.

## C    Additional Samples

DP-LDM (Ours)

DP-MEPF

$\epsilon = 10$

$\epsilon = 5$

$\epsilon = 1$

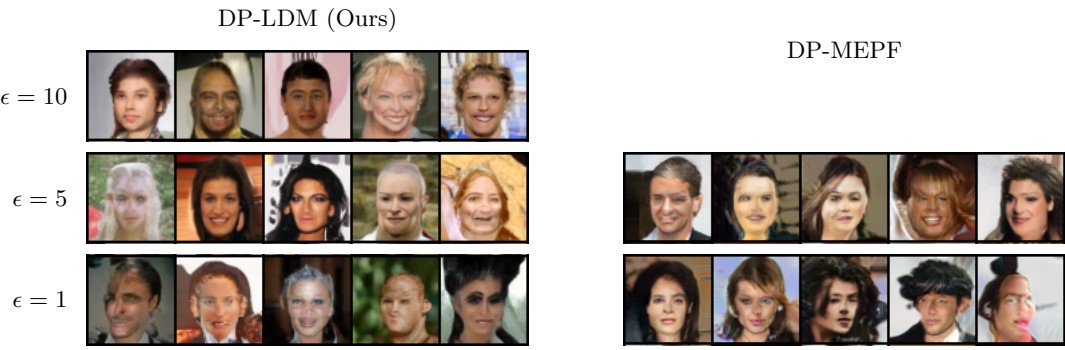

Figure 11: Synthetic $64 \times 64$ CelebA samples generated at different levels of privacy. Samples for DP-MEPF are taken from Harder et al. (2023).

DP-LDM (Ours)            DP-Diffusion

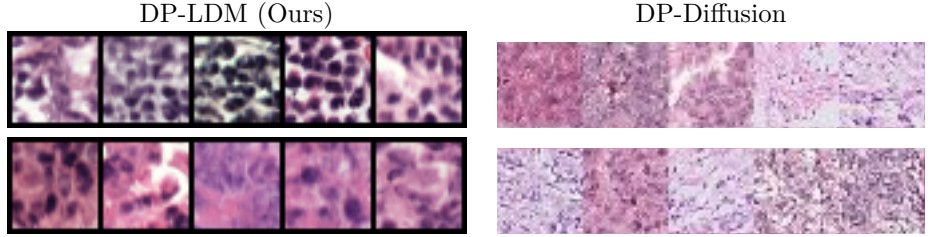

Figure 12: Synthetic $32 \times 32$ Camelyon17-WILDS samples at $\epsilon = 10$ and $\delta = 3 \times 10^{-6}$. Samples for DP-Diffusion are taken from table 1 in Ghalebikesabi et al. (2023).

## D    Comparison between DP and non-DP Samples

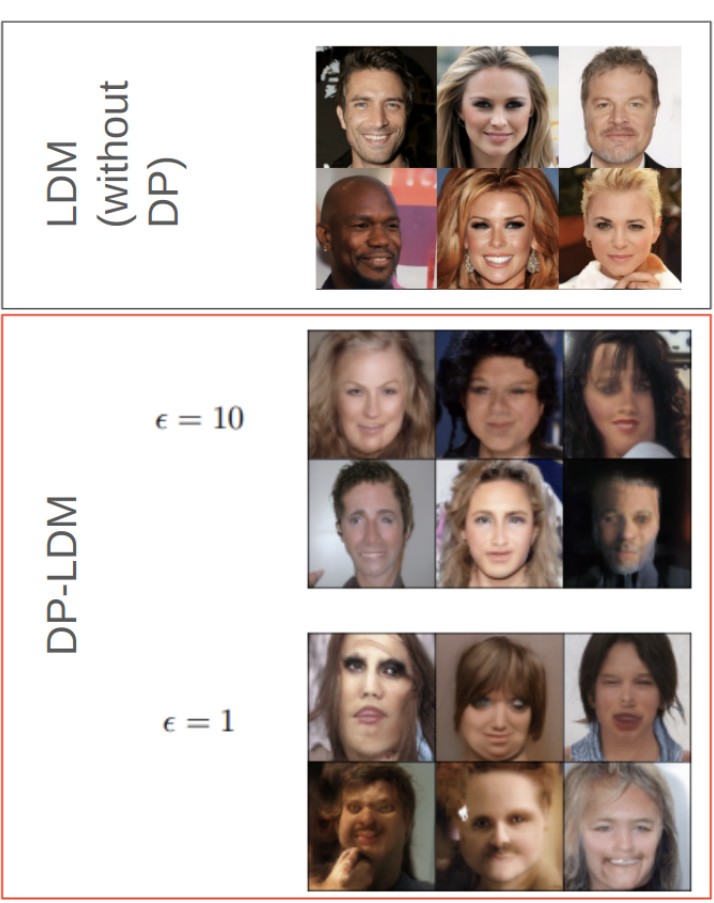

Figure 13: Synthetic CelebAHQ images from a non-DP-trained LDM (Top, black box, images taken from Rombach et al. (2022)) versus those from DP-trained LDMs (Bottom, red box). The non-DP synthetic images exhibit humanly discernible characteristics of faces. On the other hand, DP synthetic samples exhibit distortions at varying levels, where a higher privacy guarantee ($\epsilon = 1$) yields more distortion.

