# OpenReview forum: "Differentially Private Latent Diffusion Models"
_TMLR — Accepted by TMLR_

### Review · Reviewer_g9at · 2024-07-22

**Summary Of Contributions:**

The paper addresses the significant privacy concerns associated with diffusion models (DMs), which are widely used for image generation but have been shown to leak training data. The authors propose a novel approach that combines differential privacy (DP) with latent diffusion models (LDMs) to enhance privacy while maintaining image quality. By fine-tuning only the attention modules of the LDMs using differential privacy stochastic gradient descent (DP-SGD), they reduce the number of trainable parameters by approximately 90%, leading to improved efficiency and a better privacy-utility trade-off. The method allows for the generation of high-dimensional images (256x256) conditioned on text prompts, marking a first in the field of differentially private image generation. The authors demonstrate that their approach outperforms existing state-of-the-art methods on various benchmark datasets. Additionally, the reduced computational requirements make this method more accessible and environmentally friendly, democratizing the use of DP in image generation.

**Audience:**

Yes

**Broader Impact Concerns:**

Generative tools are becoming more and more realistic and of general availability, with the risk of generating harmful content, especially when generating human-related content. A Broader Impact Statement analyzing the impact of these tools is required.

**Claims And Evidence:**

Yes

**Requested Changes:**

The paper needs some revision from a clarity point of view. Additionally, I feel like the comparison with LoRA is not completely fair in the current setting.

**Strengths And Weaknesses:**

## Strengths

- ****Innovative Methodology:**** The integration of LDMs with DP-SGD represents a significant advancement in the field, allowing for efficient training while addressing privacy concerns effectively.
- ****Performance Improvement:**** The proposed DP-LDMs outperform many existing methods, showcasing their ability to generate high-quality images while maintaining differential privacy.
- ****Reduced Resource Requirements:**** By significantly lowering the number of parameters that need to be fine-tuned, the authors demonstrate a substantial decrease in GPU hours required for training, making the approach more accessible to researchers and practitioners.
- ****First Attempy:**** This work is notable for being the first to produce high-dimensional images conditioned on text prompts with DP guarantees.

## Weaknesses

My main concern regards the clarity of the paper. There are several undefined notations or paragraphs that would benefit from re-writing. Examples include the missing definition of $\tau_\theta$ in Sec. 2.1 (especially since this replaces the traditional \eps in Diffusion Models), the paragraph **Which public dataset to use given a private dataset?** where it is not clear what the authors mean by "similarity in what sense".

From the evaluation point of view, I have doubts regarding the comparison of the proposed method w.r.t. LoRA fine-tuning, as the number of parameters for LoRA is almost an order a magnitude different from the proposed one. What would be the impact of a higher rank with a similar number of parameters to the authors' proposal?

Additionally, a couple of other remarks:
- The tables captions position is not consistent across the paper.
- $\alpha$ in LoRA is not defined as $1/r$ as reported in Sec. 5.4. It is an hyperparameter whose default value is $r$ as reported in the original LoRA paper.

---

> ### Author Response · Authors · 2024-09-10
>
> **Specify the Missing definition of $\tau_\theta$ in Sec 2.1:** We added this definition below eq(1).
>
> **Rewrite "Which public dataset to use given a private dataset?", to make it clear for similarity in what sense:** We rewrote this paragraph. Please see the changes in red.
>
> **Impact of a higher rank in DP-LoRA with a similar number of parameters to DP-LDM:** The reason we showed the increasing ranks in DP-LoRA is to illustrate that the parameters of attention blocks do need high-rank approximations to perform well. If the inherent dimension of those parameters is truly low dimensional, low-rank approximation to those should perform equally well as our DP-LDM. Hence, we think this is a fair comparison between DP-LoRA and DP-LDM.
>
> **Table captions position not consistent throughout the paper:** We have updated all captions in the paper to be below their respective figures/tables (changes in red).
>
> **$\alpha$ in LoRA is not defined as $1/r$, it is a hyperparameter whose value is $r$ in LoRA.:** We thank the reviewer for pointing this out. We have updated the text to state that as in the original LoRA paper, $\alpha$ is fixed to the first rank that we try. As we begin with rank $1$ for all experiments, $\alpha$ is fixed to $1$. To confirm that $\alpha$ can be fixed to a constant value, we re-run some experiments with $\alpha$ set to $16$ as in LoRA. More specifically, we train our DP-LDM model on unconditional CelebA64 at $\epsilon=10$ with varying learning rates and ranks while keeping all other hyperparameters fixed to their values in the paper. The best result (with LoRA) we achieved in our original experiments was at a rank of $8$ and a learning rate of 1e-6, producing an FID of $15.95$. With $\alpha=16$, the best result is also at rank $8$ but with a learning rate of 5e-7, producing an FID of $16.68$. This result suggests that increasing the scaling factor alpha is similar to decreasing the learning rate.
>
> **Broader Impact Statement:** We have added a broader impact statement to the end of the main text. Thank you for the suggestion.

---

### Review · Reviewer_ofXX · 2024-07-22

**Summary Of Contributions:**

The paper introduce differential privacy to the latent diffusion models while finetuning only 10% of their parameters. The idea is simple, it shows good results, and reduces the recources needed for this task by a lot.

**Audience:**

Yes

**Claims And Evidence:**

Yes

**Requested Changes:**

I think the paper needs a refinement but this is not a major problem for me

**Strengths And Weaknesses:**

Strengths

1. Privacy is a crucial subject and this is a subject that is relevant to the readers of TMLR
2. The proposed method seems to perform quite well compared to the state of the art
3. The method dictates to finetune only the attention modules (10% of the parameters) and there is a computation time comparison with the literature
4. The paper's claims are supported by the results. Also the quantity of the experiments and ablations is satisfactory

Weaknesses

1. Although the manuscript is not bad written, it seems a bit rushed, for example there is no consistency between references (Fig. 5 vs Figure 11), or instead of "figure 3" and "figure 4" there is "figure 5.1" and "figure 5.2" which is the numbering of the subsections.

---

> ### Author Response · Authors · 2024-09-10
>
> **Inconsistent references:** Thank you for pointing this out. We have fixed the inconsistent references and marked the changes in red.

---

### Review · Reviewer_HLp2 · 2024-09-04

**Summary Of Contributions:**

This paper proposes to fine-tune the attention layer of latent diffusion models for imposing differential privacy on them. The main idea is to train a latent diffusion model on a public dataset and then fine-tune the attention layers of this latent diffusion model on the private dataset.  For this purpose, it applies the idea of DP-SGD by adding a Gaussian noise to the gradients.

**Audience:**

Yes

**Broader Impact Concerns:**

No concerns on ethical implications of the work.

**Claims And Evidence:**

No

**Requested Changes:**

- Discuss more about DP in diffusion models and generative models including what we aim to obtain and how to measure/quantify them.

- Discuss more about the difference of generated images of the standard diffusion models and DP ones. Why can you say that the generated images of DP diffusion models do not reveal the information of private data?

- Incorporate the performance metrics to measure/quantify the DP level of the proposed and baseline approaches.

**Strengths And Weaknesses:**

### Strengths
- The proposed approach is faster, while maintaining good generation quality comparing to some DP diffusion model baselines.

### Weaknesses
- The novelty is very limited because it simply combines existing techniques such as LDM, DP-SGD, and fine-tuning attention layers.
- The paper lacks the depth discussion of DP for diffusion models or generative models. Generally, we do not want the leak of private data in the diffusion models, however, we can use the diffusion models to regenerate the private data. This seems controversy to me if this is not deliberately discussed.
-  In experiments, the paper reports the FID score and accuracy of the downstream tasks, I cannot see any performance metrics about the differential privacy. Therefore, it is unable to justify if the proposed approach is more DP than others or if it can guarantee DP up to any extent.

---

> ### Author Response · Authors · 2024-09-10
>
> **What we aim to obtain and how to measure/quantify them:** We added a few lines in the first paragraph of the introduction (in Red) to emphasize what this paper aims at. In a nutshell, diffusion models memorize training data, and Carlini et al. 2023 [1] attempted to use differential privacy as a remedy for this issue. However, their vanilla application of DP-SGD to the scale of Stable Diffusion did not work (as it is expected). In our paper, we provide a practical and scalable solution to fine-tune diffusion models such that the resulting model does not produce identical images as those in the private dataset.
>
> **Generated images of standard LDM and DP-LDM. How do we interpret “privacy” in those generated images?** In Appendix D, we added Figure 13 to compare the generated images of standard LDM (top) and DP-LDM (bottom) at two different epsilon levels.  Synthetic CelebAHQ images from a non-DP-trained LDM exhibit humanly discernible characteristics of faces. On the other hand, DP synthetic samples exhibit distortions at varying levels, where a higher privacy guarantee ($\epsilon = 1$) yields more distortion.
>
> **How to measure/quantify DP levels of DP-LDM and other baseline methods:** Differential privacy is a mathematical notion whose mathematical definition provides the precise level of privacy guarantee in terms of $\epsilon$ and $\delta$. This means there is no need to quantify the privacy guarantee, as the definition provably guarantees it. So, when we compare our method to other SOTA methods, we typically set a guaranteed privacy level across all methods, then see which method yields the best performance.  For instance, in Table 2, the Top box shows FID scores (a performance metric) of all methods at the same privacy level, e.g., the first column shows the privacy level set to $\epsilon=10$ and $\delta=0.00001$, the second column shows the privacy level set to $\epsilon=5$ and $\delta=0.00001$. Since our FID score is lower than others at these privacy levels, we conclude DP-LDM outperforms others, i.e., achieving higher utility at the same privacy guarantee.
>
> [1] Nicholas Carlini, Jamie Hayes, Milad Nasr, Matthew Jagielski, Vikash Sehwag, Florian Tramèr, Borja Balle, Daphne Ippolito, and Eric Wallace. Extracting training data from diffusion models. In Proceedings of the 32nd USENIX Conference on Security Symposium, SEC ’23, USA, 2023. USENIX Association. ISBN 978-1-939133-37-3.

---

### Decision · Action_Editor_pRE3 · 2024-10-17

**Recommendation:** Accept with minor revision

**Comment:**

Two of our three reviewers recommended acceptance, and the AE agrees with the recommendations that the benefits of publishing this paper in our community outweigh any downsides. The AE has made some recommendations to mitigate these downsides in future research and looks forward to seeing them addressed in the camera-ready version.

Limitation section

I encourage the authors to have the limitation section at the end of the main body.

>  While the paper proposes a technique to counterattack the privacy risks of diffusion models, the evaluation has not been done against existing privacy attacks. So we know that the diffusion model is private "under the definition of DP," but we do not know whether this is sufficient to mitigate our privacy concerns.

> I would also emphasize that the definition of privacy within the community has not yet reached a consensus. Carlini et al.'s work is a single example of exploitation, but it remains unclear what aspects we aim to protect in generated data. For instance, should we expect a private model to avoid generating a small pimple on my left nostril, for example? Such an unclear definition may expose practitioners employing this private training mechanism to the "false sense of privacy"---they thought that they were private but turned out not to be.

Presentations

1. I would remove "which, to the best of our knowledge, has not been attempted before," as the implications are unclear.
2. I would revise the paragraph starting with "Great performance..." in the introduction. It's unclear what it means to be "great" when generating photos from private diffusion models.
3. Figures and Tables: The current structure of tables and figures does not look professional—the fonts are too small, or the photos are too small to identify and compare the differences. The placements are sometimes off from the text explaining figures and tables. I would fix them.
4. In Broader Impact Statement: "To guarantee the reproducibility" -> "To encourage the reproducible research practices,"

**Audience:**

The paper presents one baseline approach to making diffusion models private, which can be a countermeasure for someone concerned about data extraction attacks on diffusion models.

**Claims And Evidence:**

The key contribution this paper claims is the adaptation of differentially-private model training (DP-SGD) to a new class of models: diffusion models. The primary challenge lies in the scale of these diffusion models, which makes training with differential privacy computationally intractable. The technical approach this paper takes makes sense: by introducing a dimensionality reduction technique and focusing on training a substructure of the models. The evaluation shows high-resolution images "efficiently" generated from diffusion models trained with privacy guarantees and compares their quality.